# Clinical characteristics, risk factors, and pathogen spectrum of mixed infections in children with *Mycoplasma pneumoniae* pneumonia: a retrospective cohort study of 1,428 cases from a children's medical centre in East China

Cheng Wang,[1] Shenghao Hua,[1] Yang Li,[1,2] Yuanyuan Gao,[1] Lunjing Yan,[1] Xuejun Shao,[1,2] Jun Xu,[1] Xin Zhang[1,2]

**ABSTRACT** This study aimed to investigate the clinical characteristics, risk factors, and pathogen spectrum of mixed infections in children with *Mycoplasma pneumoniae* pneumonia (MPP). A total of 1,428 pediatric MPP patients were retrospectively included and categorized into a single infection group (*n* = 528) and a mixed infection group (*n* = 900), with the latter further divided into bacterial, bacterial–viral, and viral mixed infection subgroups. The results revealed that the children in the mixed infection group were younger (median 3–6 years vs 8 years, *P* < 0.001) and had shorter hospital stays (8 days vs 9 days, *P* < 0.001) but had a higher incidence of severe pneumonia (30.44% vs 20.27%, *P* < 0.001). Mixed infections occurred more frequently in autumn and presented with more prominent upper respiratory symptoms. Significant differences in the levels of some immune, coagulation, and inflammatory markers were detected between the two groups (*P* < 0.05). The mixed infection group had higher rates of gastrointestinal dysfunction and myocardial injury. Multivariate regression analysis revealed longer cough duration, younger age, autumn onset, rhinorrhea, decreased complement C4 levels, and elevated antithrombin III (AT-III) levels as independent risk factors for mixed infection. Pathogen spectrum analysis revealed that *Human rhinovirus* and *Streptococcus pneumoniae* were the predominant viral and bacterial pathogens, respectively, in mixed infections. Mixed infections accounted for a substantial proportion of the MPP cases (63.03%) and were associated with a higher risk of severe disease. It is recommended that active pathogen testing be conducted in children with MPP and that a prediction model incorporating age, season, symptoms, and laboratory indicators be developed for early identification and precise intervention.

**IMPORTANCE** This study highlights the importance of mixed infections in pediatric MPP patients. The results indicate that mixed infections account for up to 63.03% of MPP cases in children and are significantly associated with an increased risk of severe pneumonia. The key risk factors identified include younger age, autumn onset, rhinorrhea, decreased complement C4 levels, and elevated AT-III levels, providing a basis for the early identification of high-risk patients. Pathogen spectrum analysis identified *human rhinovirus* and *Streptococcus pneumoniae* as the predominant mixed infecting pathogens. These findings suggest that multipathogen testing should be implemented for children with MPP. This research supports the development of a risk prediction model to enable early warning and targeted intervention, thereby optimizing treatment, improving prognosis, and enhancing public health prevention and control strategies.

**KEYWORDS** *Mycoplasma pneumoniae*, mixed infection, children, risk factors, prediction model, complement C4, pathogen spectrum

**Peer Reviewers** Xinxing Zhang, Sichuan University, Chengdu, China; Yue Tao, Shanghai Children's Medical Center, Shanghai Jiaotong University School of Medicine, Shanghai, China; Qing Cao, Shanghai Children's Medical Center, Shanghai Jiaotong University School of Medicine, Shanghai, China

Address correspondence to Xin Zhang, wyc1116@foxmail.com, Jun Xu, 13706207528@139.com, or Xuejun Shao, xjshao@suda.edu.cn.

Cheng Wang, Shenghao Hua, and Yang Li contributed equally to this article. Author order was determined both alphabetically and in order of increasing seniority.

The authors declare no conflict of interest.

See the funding table on p. 18.

*M*ycoplasma pneumoniae (MP) is among the important pathogens of community-acquired pneumonia (CAP) in children and is especially common among school-aged children. A review study from the *Journal of the Korean Medical Association* revealed that MP has a typical 3- to 5-year epidemic cycle, with a notably significant increase in infection rates observed globally since mid-2023 (1). This post-COVID-19 pandemic surge is closely linked to the "immune debt" phenomenon following the relaxation of containment measures. As a group of small prokaryotic microorganisms lacking a cell wall structure, MP causes damage not only through direct adhesion to respiratory epithelial cells via its adhesins and related accessory proteins but also, more importantly, by triggering a complex host immune–inflammatory response. This process often leads to respiratory and systemic multisystem complications (2–4).

Notably, the clinical manifestations of *Mycoplasma pneumoniae* pneumonia (MPP) range from mild, self-limiting respiratory infections to severe pneumonia requiring hospitalization. Particularly concerning is the increasing proportion of refractory *Mycoplasma pneumoniae pneumonia* (RMPP) cases (5–7). RMPP is characterized by persistent fever, worsening clinical symptoms, and progressive radiological lung findings despite adequate macrolide antibiotic therapy for more than a week, potentially progressing to severe complications such as pulmonary necrosis and pleural effusion (8, 9). Studies suggest that an immune-mediated hyperinflammatory response is central to the pathogenesis of RMPP and that mixed infections may be key triggers of this excessive immune response (10).

Indeed, the incidence of mixed infection in children with MPP is considerably high, with recent studies reporting rates of up to 63.03%, highlighting the necessity and urgency of in-depth research into mixed MPP infections. In clinical practice, the presence of mixed infections significantly alters the disease course and treatment response in patients with pediatric MPP, making it a major challenge in the field of pediatric respiratory infections. A 2024 review discussing the global increase in the incidence and severity of MPP after the COVID-19 pandemic revealed that the increase in mixed infections was a key factor contributing to more severe disease and complex clinical management (11). Mixed infection refers to simultaneous or sequential infection with MP and other pathogens. Although mixed MP infection is common, its pathogen spectrum is not homogeneous but is strongly influenced by the regional epidemiological background. International data show that the copathogen spectrum of MPP in America is often closely coupled with seasonal respiratory viruses (such as such as human metapneumovirus [HRV] and respiratory syncytial virus [RSV]) and bacterial (such as *S. pneumoniae*) epidemic cycles (12). Some studies have also revealed significant differences between North and South China: pathogen investigations in North China (such as Beijing) highlight the high proportion of MP in the overall pathogen composition of community-acquired pneumonia (13). However, research in the southern region of China focuses more on the molecular epidemiological characteristics of MP itself, suggesting possible differences in epidemic genotypes (14). Studies have shown that the resistance rate of pediatric infection strains in the Hebei region to macrolide drugs is still high, which not only leads to atypical clinical manifestations and treatment difficulties but also may affect the susceptibility and pathogen combination patterns of mixed infections by altering the host immune response and disease course (15). This infection pattern can significantly modify the clinical presentation, severity, and treatment response of MPP. A 2025 multicenter study by Yu et al. (8) on severe MPP with liver injury revealed that children with mixed infections often presented with more severe pulmonary manifestations and a higher incidence of liver injury. These patients experience longer hospital stays, higher health care costs, and greater needs for intensive care and oxygen support.

Despite the widely recognized clinical importance of mixed infections in MPP, research on their specific pathogen spectrum composition, risk factors, and characteristic laboratory indicator changes remains insufficient. There is a particular lack of large-scale epidemiological studies revealing the seasonal variation patterns and pathogen combination modes of mixed infections. Furthermore, the early identification and risk

prediction of mixed infections still lack reliable and specific biomarkers. To address these research gaps, the aim of this study was to thoroughly investigate the clinical characteristics, independent risk factors, and epidemiological features of the pathogen spectrum in mixed infections through a retrospective analysis of clinical data from 1,428 pediatric MPP patients. The specific research objectives included comparing the clinical features, laboratory indicators, and complications between children with simple MP infections and those with mixed infections using multivariate logistic regression analysis; establishing a receiver operating characteristic (ROC) curve for key predictive indicators to determine their optimal cutoff values; and analyzing the pathogen spectrum composition of mixed infections and their seasonal variation patterns. The findings of this study provide important evidence for early clinical diagnosis, precise treatment, and risk assessment of mixed infections in pediatric MPP patients, with the potential to improve patient outcomes and reduce the risk of severe complications.

## MATERIALS AND METHODS

### Study population and grouping

#### Inclusion criteria

(i) aged 0–16 years; (ii) hospitalized between July 2021 and July 2023; (iii) diagnosed with MPP according to *Zhu Futang's Practical Pediatrics (8th Edition)* (16) and the Chinese Society of Pediatrics Guidelines for CAP in Children (2023) (17): presence of acute respiratory symptoms and radiological evidence of pneumonia (chest X-ray/CT showing infiltration), plus positive MP detection (MP-DNA via polymerase chain reaction [PCR] or MP-IgM via chemiluminescence). The criteria for assessing the severity of pneumonia in children has been previously published (18). Research location: As the largest tertiary Grade A children's specialized hospital in the local area, the *Children's Hospital of Soochow University* covers relatively severe cases in Suzhou and its surrounding areas. Therefore, the results of this study reflect mainly the epidemiological and clinical characteristics of mixed MP infection in hospitalized children from local areas.

#### Exclusion criteria

(i) Foreign body aspiration; (ii) readmission within 1 month; (iii) incomplete clinical data; and (iv) congenital heart disease, immunodeficiency, or malignant tumors.

#### Grouping

Patients were divided into two groups on the basis of the pathogen detection results: (i) the single infection group, in which only MP was detected (negative for other respiratory pathogens via 13-plex nucleic acid testing and bacterial culture) and (ii) the mixed infection group, in which MP was detected and combined with at least one other pathogen, including the mixed bacterial infection group, mixed bacterial-viral infection group, and mixed viral infection group.

### Data collection

(i) Clinical data were extracted from the hospital's electronic medical records, including demographic data such as age, sex, delivery mode (vaginal/cesarean), and premature birth history; (ii) clinical characteristics such as season onset (spring: Mar-May; summer: Jun-Aug; autumn: Sep-Nov; winter: Dec-Feb); symptoms (fever grade: low-grade <38°C, moderate 38–39°C, high-grade >39°C; nasal congestion, rhinorrhea, wheezing, tachypnea, three depression sign, vomiting); physical signs (rhonchi/moist rales, wheezing, rales, and decreased breath sounds); and radiological findings (pleural effusion, atelectasis, and pulmonary consolidation); and (iii) laboratory indicators such as immune indicators (C3, C4, Ig, CD3+%, CD4+/CD8+, etc.), inflammatory indicators (PCT,

SAA, etc.), and coagulation function indicators (PT, APTT, etc.). Complications included gastrointestinal dysfunction, myocardial damage, and hypokalemia.

## Laboratory testing

### Pathogen detection

Deep sputum or bronchoalveolar lavage fluid (1–5 mL) was collected for testing. (i) Nucleic acid was extracted using an automated extraction system (Jiangsu Shuoshi Biotech). Thirteen-plex respiratory pathogen nucleic acid testing was performed using a respiratory pathogen multiplex PCR kit (Haiershi Gene Technology Co., Ltd; Ningbo; China), targeting 13 pathogens, namely, MP, *human rhinovirus* (HRV), *respiratory syncytial virus* (RSV), *human parainfluenza virus* (HPIV), *human coronavirus* (HCoV), *human metapneumovirus* (HMPV), *human bocavirus* (HBoV), *human adenovirus* (HADV), *chlamydia* (Ch), and *influenza virus* (InfA H1/H3, InfB), by PCR. Analyses were performed on a 3500XL-Dx Genetic Analyzer. (ii) MP-IgM antibody testing: Serum samples were tested for MP-IgM antibodies using an *M. pneumoniae* IgM Antibody Detection Kit (Chemiluminescence Immunoassay; YHLO Biotechnology, Shenzhen, China) on a YHLO iFlash 3000 chemiluminescence analyzer. In accordance with the manufacturer's specifications, the assay has a sensitivity of ≥90% and a specificity of ≥85%. A result was considered positive when the cutoff index (COI) was ≥1.0. (iii) Bacterial culture: Specimens were inoculated on blood agar and chocolate agar plates (Antu Biotechnology Co., Ltd; Zhengzhou; China) and incubated at 37°C with 5% $CO_2$ for 24–48 h. Qualified sputum was assessed using <10 squamous epithelial cells/low-magnification field and >25 white blood cells/low-magnification field. Sputum specimens were examined using the four part streaking method. The main pathogens were *S. pneumoniae*, *H. influenzae*, *M. catarrhalis*, and *S. aureus* (usually considered positive). Even if the growth was 1+ or 2+, as long as the following bacteria were isolated, the result was usually considered clinically significant; gram-negative bacteria (such as *K. pneumoniae*, *P. aeruginosa*, and *A. baumannii*) were required to be combined with the growth rate, whereas 3+ or 4+ growth was considered meaningful. Bacterial identification was performed using a Microflex LT/SH Automated Mass Spectrometer (Bruker Daltonics, Bremen, Germany). Direct selection of single bacterial colonies from the plate for testing could yield species-level identification results within minutes, greatly optimizing the identification process of the sputum culture. The confidence score (e.g., >90%) was used to determine whether the result was reliable.

### Blood biomarker detection

Venous blood (3–5 mL) was collected. Please refer to Table 1 for details of the detection methods.

## Statistical analysis

Data extraction was repeated by trained personnel to ensure accuracy. Blinding was not used during the data analysis process. The data were analyzed using SPSS 27.0. To test for normality test, the Shapiro−Wilk test for continuous data was used. Continuous data are expressed as the mean ± standard deviation and were compared via an independent samples *t*-test; nonnormally distributed data are expressed as the median (interquartile range, IQR) and were compared via the Mann−Whitney *U* test. Categorical data are expressed as the *n* (%) and were compared via *the chi-square test*. The level of complement C4 was originally a continuous variable. To facilitate clinical stratification and analyze its nonlinear relationship with outcomes, it was converted into a categorical variable based on the quartiles of the population sample. A two-tailed *P* < 0.05 was considered to indicate statistical significance.

**TABLE 1** Methods for detecting blood biomarkers

| Indicator/reference range | Method | Reagent/instrument |
|---|---|---|
| C3 (0.9–1.8 g/L), C4 (0.16–0.38 g/L), IgA (0.38–2.22 g/L), IgG (5.9–14.3 g/L), IgM (0.48–2.08 g/L) | Immunoturbidimetry assays | C3/C4/Ig Kit (Meikang Biotechnology Co., Ltd, Ningbo, China); Siemens ADVIA 2400 Biochemical Analyzer |
| Lymphocyte subsets (CD3+ (55%–83%), CD3+CD4+ (28%–57%), CD3+CD8+ (10%–39%), CD4+/CD8+ (0.98–1.94), CD3−CD19+ (6%–19%), CD3−CD(16 + 56)+ (7%–31%), CD19+CD23+ (3.8%–9.7%) | Flow cytometric analysis | Fluorescent Monoclonal Antibody Kit (Tongsheng Era Biotechnology Co., Ltd; Beijing; China); BD FACS Canto II Cytometer |
| Coagulation parameters (PT (9.8–12.1 s), APTT (21.1–36.5 s), Fib (1.8–3.5 g/L), TT (14–21 s), AT-III (80%–120%), DD-PLUS (0–550 µg/L), FDP [0–5,000 µg/L]) | Coagulation assay (AT-III: chromogenic substrate; DD-PLUS, FDP: Immunoturbidimetric assays) | Coagulation Kit (Hisense Medical Electronics Co., Ltd, Shanghai, China); CS-5100 Fully Automated Coagulation Analyzer |
| Procalcitonin (PCT [≤0.5 ng/mL]) | Electrochemiluminescence immunoassay | PCT Kit (Roche Technology Co., Ltd, Shanghai, China); Cobas 6000 Analyzer |
| Heparin-Binding Protein (HBP [0–11.4 ng/mL]) | Immunofluorescence dry quantitative method | HBP Kit (Zhonghan Shengtai Technology Biotechnology Co., Ltd; Hangzhou; China); JOINSTAR JET-ISTAR 3000 Immunoanalyzer |
| Serum Amyloid A (SAA [0–10 mg/L]) | Colloidal gold method | SAA Kit (AOPU Biopharmaceutical Co., Ltd; Shanghai; China); Gold Immunochromatography Digital Quantitative Analyzer |
| Hypersensitive C-reactive protein (sCRP [0–8 mg/L]) | Immunoturbidimetry assays | sCRP Kit (Wako Pure Chemical Industries, Ltd, Japan); LABOSPECT 008α Analyzer |

## RESULTS

### Baseline characteristics of the patients

A total of 1,428 patients with MPP admitted to CHSU from July 2021 to July 2023 were enrolled in the study, including 745 males (52.17%) and 683 females (47.83%), with a male-to-female ratio of 1.09:1. The median patient age was 5 years (IQR: 2–8 years). The mixed infection group accounted for 63.03% (900/1428) of the cases, including 185 cases of mixed bacterial infections, 285 cases of mixed bacterial viral infections, and 430 cases of mixed viral infections.

### Clinical characteristics

There were no significant differences in sex, premature birth history, or delivery mode between the single infection group and the mixed infection group (all $P > 0.05$; Table 2). The cough duration was longer in the mixed infection group, but the hospitalization time was shorter, and the median age (4 years) was significantly younger in the mixed infection group than in the single infection group (8 years) ($P < 0.05$). The mixed infection group had a greater proportion of patients with a history of pneumonia and wheezing, a longer cough duration, and higher frequencies of low-grade fever, nasal congestion, rhinorrhea, the three depression sign, vomiting, and wheezing/rales (all $P < 0.05$). The analysis of differences between the single infection group and the mixed infection

**TABLE 2** Comparison of clinical characteristics between single infection group and mixed infection group[a]

| Clinical characteristics | Single infection group (n = 528) | Mixed infection group (n = 900) | t/Z/χ2 | P |
|---|---|---|---|---|
| Hospital stay (days, IQR) | 9 (7–10) | 8 (7–9) | −4.578 | **<0.001** |
| Cough duration (days, IQR) | 6.07 ± 3.47 | 7.64 ± 6.67 | −5.023 | **<0.001** |
| Age (years, IQR) | 8 (6–9) | 4 (2–7) | −14.892 | **<0.001** |
| Male, n (%) | 266 (50.38%) | 478 (53.11%) | 0.995 | 0.318 |
| Premature delivery, n (%) | 20 (3.79%) | 51 (5.67%) | 2.486 | 0.115 |
| Vaginal delivery, n (%) | 311 (58.90%) | 545 (60.56%) | 0.379 | 0.538 |
| Severe pneumonia, n (%) | 107 (20.27%) | 274 (30.44%) | 17.627 | **<0.001** |
| History of pneumonia, n (%) | 62 (11.74%) | 143 (15.89%) | 4.654 | **0.031** |
| History of wheezing, n (%) | 31 (5.87%) | 91 (10.11%) | 7.656 | **0.006** |
| Onset season, n (%) | | | | |
| Spring | 163 (30.87%) | 265 (29.44%) | 0.323 | 0.570 |
| Summer | 171 (32.39%) | 249 (27.67%) | 3.570 | 0.059 |
| Autumn | 96 (18.18%) | 254 (28.22%) | 18.131 | **<0.001** |
| Winter | 98 (18.56%) | 132 (14.67%) | 3.734 | 0.053 |
| Fever grade, n (%) | | | | |
| Low fever (<38°C) | 9 (1.70%) | 46 (5.11%) | 10.428 | **0.001** |
| Moderate fever (38–39°C) | 156 (29.55%) | 308 (34.22%) | 3.318 | 0.069 |
| High fever (>39°C) | 346 (65.53%) | 444 (49.33%) | 35.320 | **<0.001** |
| Symptoms, n (%) | | | | |
| Nasal congestion | 141 (26.70%) | 418 (46.44%) | 54.433 | **<0.001** |
| Rhinorrhea | 129 (24.43%) | 468 (52.00%) | 103.955 | **<0.001** |
| Three depression sign | 7 (1.33%) | 42 (4.67%) | 11.209 | **0.001** |
| Vomiting | 98 (18.56%) | 255 (28.33%) | 17.079 | **<0.001** |
| Lung signs, n (%) | | | | |
| Rhonchi rales | 33 (6.25%) | 48 (5.33%) | 0.523 | 0.470 |
| Moist/rales | 422 (79.92%) | 677 (75.22%) | 4.149 | **0.042** |
| Wheezes/rales | 81 (15.34%) | 271 (30.11%) | 39.086 | **<0.001** |
| Decreased breath sound | 116 (21.97%) | 107 (11.89%) | 25.663 | **<0.001** |
| Radiological findings, n (%) | | | | |
| Atelectasis | 28 (5.30%) | 32 (3.56%) | 2.525 | 0.112 |
| Pulmonary consolidation | 50 (9.47%) | 38 (4.22%) | 15.846 | **<0.001** |

[a]Index data (Hospital stay, Age) are represented by median (IQR) and analyzed using *Mann Whitney U test*. Index data of Cough duration is represented by mean ± standard deviation and analyzed using *t-test*. Other count data are expressed as percentages and analyzed using $\chi^2$ test. The bold column of *P*-values represents meaningful data.

groups (bacterial, bacterial–viral, and viral mixed infection subgroups) is detailed in Table S1 to S3.

## Laboratory examination

As shown in Table 3, the levels of C3, C4, IgA, and IgG and the CD3+CD8+, CD3−CD(16 + 56)+, PT, Fib, DD-PLUS, SAA, and sCRP cell ratios were greater in children with single infections than in those with mixed infections (all *P* < 0.05). Conversely, in the single infection group, the levels of IgM and the CD4+/CD8+, CD3−CD19+, and AT-III cell ratios were lower than those in the mixed infection group (all *P* < 0.05). CD3+%, CD3+CD4+%, CD19+CD23+%, APTT, TT, FDP, PCT, and HBP did not significantly differ between the groups (all *P* > 0.05). The analysis of differences between the single infection group and the mixed infection groups (bacterial, bacterial–viral, and viral mixed infection subgroups) is detailed in Tables S4 to S6.

## Comparison of complications

As shown in Table 4, compared with the single infection group, the mixed infection group had significantly greater incidences of gastrointestinal dysfunction (11.77% vs

**TABLE 3** Comparison of laboratory indicators between single infection group and mixed infection group[a]

| Lab test | Single infection group (n = 528) | Mixed infection group (n = 900) | t/Z | P |
|---|---|---|---|---|
| C3 (g/L) | 1.28 (1.14–1.42) | 1.23 (1.09–1.37) | −4.076 | **<0.001** |
| C4 (g/L) | 0.42 ± 0.12 | 0.38 ± 0.12 | 6.470 | **<0.001** |
| IgA (g/L) | 1.40 ± 0.67 | 1.08 ± 0.68 | 8.375 | **<0.001** |
| IgG (g/L) | 9.18 (7.72–10.90) | 8.42 (6.95–10) | −6.130 | **<0.001** |
| IgM (g/L) | 1.19 (0.93–1.48) | 1.28 (0.96–1.68) | −3.118 | **0.002** |
| CD3+ (%) | 68.55 (62.78–73.53) | 67.79 (60.78–73.58) | −1.829 | 0.067 |
| CD3+CD4+ (%) | 36.66 ± 8.19 | 37.18 ± 8.40 | −1.019 | 0.309 |
| CD3+CD8+ (%) | 25.38 ± 5.99 | 24.31 ± 7.09 | 2.682 | **0.007** |
| CD4+/CD8+ (%) | 1.48 (1.13–1.88) | 1.54 (1.20–1.98) | −2.579 | **0.010** |
| CD3−CD19+ (%) | 19.03 ± 6.79 | 21.58 ± 8.90 | −5.377 | **<0.001** |
| CD3−CD(16 + 56)+ (%) | 10.89 (6.76–16.33) | 9.49 (6.10–14.31) | −3.343 | **0.001** |
| CD19+CD23+ (%) | 5.80 (4.03–8.59) | 6.35 (4–9.70) | −1.727 | 0.084 |
| PT (s) | 13.49 ± 0.77 | 13.27 ± 0.89 | 4.614 | **<0.001** |
| APTT (s) | 39.37 ± 5.68 | 39.47 ± 6.08 | −0.268 | 0.789 |
| Fib (g/L) | 4.80 (4.20–5.29) | 4.44 (3.73–5.09) | −5.887 | **<0.001** |
| TT (s) | 15.83 ± 0.95 | 15.91 ± 1.07 | −1.348 | 0.178 |
| AT-III (%) | 111.35 ± 10.99 | 113.65 ± 12.18 | −3.393 | **0.001** |
| DD-PLUS (µg/L) | 580 (410–840) | 460 (320–697.50) | −6.433 | **<0.001** |
| FDP (µg/L) | 2,370 (1,792.50–3,117.50) | 2,230 (1,690–3,027.50) | −1.739 | 0.082 |
| PCT (ng/mL) | 0.11 (0.07–0.15) | 0.10 (0.06–0.20) | −0.807 | 0.420 |
| SAA (mg/L) | 127.10 (54.90–235) | 64.75 (21.73–185) | −2.607 | **0.009** |
| HBP (ng/mL) | 39.72 (18.21–79.73) | 41.58 (18.17–73.21) | −0.241 | 0.809 |
| sCRP (mg/L) | 13.12 (6.35–25.96) | 8.97 (2.94–21.61) | −5.268 | **<0.001** |

[a]Non normal distribution index data (C3, IgG, IgM, CD3+, CD4+/CD8+, CD3−CD (16+56)+, CD19+CD23+, Fib, DD-PLUS, FDP, PCT, SAA, HBP, sCRP) are represented by median (IQR) and analyzed using *Mann Whitney U test*. Other indicators of normal distribution data are expressed as mean ± standard deviation, and *t-test* is used for statistical analysis. The bold column of *P*-values represents meaningful data.

9.85%, $P < 0.001$) and myocardial damage (2.67% vs 0.95%, $P = 0.026$). In contrast, the single infection group had higher incidences of hypokalemia (14.39% vs 9.00%, $P = 0.002$). No significant differences in sepsis, atrial septal defect, coagulation disorders, abnormal liver function, neutropenia, bronchitis obliterans, or bronchial asthma were detected between the groups (all $P > 0.05$).

## Independent risk factors for mixed infection

Univariate logistic regression revealed seven variables with $P < 0.05$: cough duration, age, onset season (autumn), rhinorrhea, vomiting, C4 (Q4), and AT-III (Table 5). Multivariate logistic regression revealed that the independent risk factors for mixed infection were longer cough duration (OR = 1.222; $P = 0.036$), onset season (autumn) (OR = 4.760; $P = 0.033$), rhinorrhea (OR = 7.917; $P = 0.030$), and higher AT-III levels (OR = 1.070; $P = 0.029$). The independent protective factors were older age (OR = 0.609; $P < 0.001$), vomiting (OR = 0.155; $P = 0.028$), and higher C4 levels, especially in C4 (Q4) (OR = 0.020; $P = 0.002$).

## ROC curve analysis of key laboratory indicators

ROC curves were constructed for AT-III and C4 (Fig. 1). The area under the curve (AUC) for AT-III was 0.570 (95%CI: 0.537–0.602; $P < 0.001$), with an optimal cutoff of 113.50%. The AUC for C4 was 0.617 (95%CI: 0.585–0.650; $P < 0.001$), with an optimal cutoff of 0.38 g/L. Elevated AT-III indicated hypercoagulability and inflammation activation, with high specificity; elevated C4 reflected sustained activation of complement and showed good sensitivity. The details are provided in Table 6.

## Epidemiological characteristics of mixed viral infections

A total of 882 viral strains were detected in 715 children with mixed infections (79.44% of the mixed infection group). The most common viruses were HRV (401 strains, 45.46%), followed by HRSV (147 strains, 16.67%) and HPIV (87 strains, 9.86%) (Table 7). Seasonal variations were observed for six viruses (all $P < 0.05$). HRSV was highest in spring (70/147, 47.62%) and winter (22/147, 15.00%) and lowest in summer (21/147, 14.29%). HPIV was highest in summer (46/87, 52.87%) and lowest in spring (5/87, 5.75%), and HCoV was concentrated in summer (50/68, 73.53%). HMPV was highest in autumn (14/42, 33.33%) and winter (14/42, 33.33%). InfA H1N1 was highest in spring (15/16, 93.75%). InfB was detected only in the winter (7/8, 87.50%). HRV, HBoV, HADV, and Ch showed no significant seasonal variation (all $P > 0.05$).

## Vira mixed infection types

Among 715 children with mixed viral infection, 79.58% (569/715) had mixed single viral infection, 17.90% (128/715) had mixed double viral infection, 2.24% (16/715) had mixed triple viral infection, and only 0.14% (1/715) had mixed quadruple/quintuple viral infection (Fig. 2). The most common single viral mixed infection was MP + HRV (318/569, 55.89%), followed by MP + HRSV (98/569, 8.79%) and MP + HPIV (50/569, 8.44%) (Fig. 3A). The most common mixed viral infections were MP + HRV + HPIV (20/128, 15.63%) and MP + HRV + HRSV (14/128, 10.94%) (Fig. 3C).

## Epidemiological characteristics of mixed bacterial infections

A total of 476 bacterial strains were detected in 470 children with mixed infections. Six of them were simultaneously detected with *S. pneumoniae* and *H. influenzae*. Gram-positive bacteria accounted for 75.42% (359/476), with *S. pneumoniae* being the most common (306/476, 64.29%), followed by *S. aureus* (49/476, 10.29%). Gram-negative bacteria accounted for 24.58% (117/476) of the bacteria, with *H. influenzae* being the most common (90/476, 18.91%) (Table 8). The number of detected bacterial strains significantly increased annually from 2021 to 2023 (125→157→194). *S. aureus* peaked in 2022 (23/49, 46.94%), which was significantly higher than its peaks in 2021 (10/49, 20.41%) and 2023 (16/49, 32.65%). Except for winter, the seasonal incidence of *S. pneumoniae* far exceeds that of *H. influenzae* and *S. aureus*. The seasonal distribution of the top three bacteria detected is shown in Fig. 4.

TABLE 4  Comparison of complications between single infection group and mixed infection group[a]

| Complications | Single infection group ($n = 528$) | Mixed infection group ($n = 900$) | $\chi2$ | $P$ |
|---|---|---|---|---|
| Sepsis | 1 (0.19) | 9 (1.00) | 3.144 | 0.076 |
| Atrial septal defect | 1 (0.19) | 7 (0.78) | 2.068 | 0.150 |
| Gastrointestinal dysfunction | 52 (9.85) | 106 (11.78) | 14.981 | **<0.001** |
| Coagulation disorders | 3 (5.68) | 5 (5.56) | 0.001 | 0.975 |
| Abnormal liver function | 8 (1.52) | 15 (1.67) | 0.048 | 0.826 |
| Hypokalemia | 76 (14.39) | 81 (9.00) | 9.894 | **0.002** |
| Hyponatremia | 18 (3.41) | 27 (3.00) | 0.302 | 0.669 |
| Myocardial damage | 5 (0.95) | 24 (2.67) | 4.946 | **0.026** |
| Systemic inflammation | 8 (1.52) | 18 (2.00) | 0.438 | 0.508 |
| Neutropenia | 2 (3.79) | 11 (1.22) | 2.624 | 0.105 |
| Bronchitis obliterans | 1 (0.19) | 1 (0.11) | 0.146 | 0.703 |
| Bronchial asthma | 11 (2.08) | 13 (1.44) | 0.822 | 0.365 |

[a]Bold column of *P*-values represents meaningful data.

**TABLE 5** Multivariate logistic regression analysis of risk factors for mixed infection[a]

| Variable | B | SE | Wald $\chi2$ | P | OR (95% CI) |
|---|---|---|---|---|---|
| Cough duration | 0.200 | 0.095 | 4.412 | 0.036 | 1.222 (1.013–1.473) |
| Age | −0.497 | 0.139 | 12.733 | <0.001 | 0.609 (0.463–0.799) |
| Autumn | 1.560 | 0.731 | 4.557 | 0.033 | 4.760 (1.136–19.939) |
| Rhinorrhea | 2.069 | 0.953 | 4.713 | 0.030 | 7.917 (1.223–51.266) |
| Vomiting | −1.867 | 0.850 | 4.822 | 0.028 | 0.155 (0.029–0.818) |
| C4 (Q4) | −3.915 | 1.282 | 9.319 | 0.002 | 0.020 (0.002–0.246) |
| AT-III | 0.068 | 0.031 | 4.766 | 0.029 | 1.070 (1.007–1.138) |

[a]C4 is divided into a quartile array: low value group Q1 ($<P_{25}$), medium low value group Q2 ($P_{25}$-$P_{50}$), medium high value group Q3 ($P_{50}$-$P_{75}$), and high-value group Q4 ($>P_{75}$) The C4 level in the high-value group Q4 is an independent factor that distinguishes between single MP infection and mixed infection.

## DISCUSSION

MPP is a leading cause of CAP in children aged 3–15 years (19), and mixed infections with viruses or bacteria have become key factors that aggravate disease severity and complicate clinical management (20–23). Compared with single MPP infections, mixed infections are associated with more severe clinical symptoms and a higher incidence of complications (22, 23). Given the regional and seasonal variability of pathogen spectra in MPP mixed infections, large-sample epidemiological and mechanistic studies are urgently needed to guide clinical practice. This retrospective cohort study of 1,428 children with MPP systematically explored the epidemiological characteristics, independent risk factors, pathogen spectrum patterns, and changes in immune-coagulation function of mixed infections. The core findings included the following: mixed infections accounted for 63.03% of the MPP cases; younger age, autumn onset, and rhinorrhea were key risk factors; complement C4 reduction and AT-III elevation served as potential predictive biomarkers; HRV and *S. pneumoniae* were the predominant coinfecting pathogens; and mixed infections were associated with higher risks of severe pneumonia, gastrointestinal dysfunction, and myocardial damage. In the next section, we discuss these findings together with pathogenic mechanisms and the literature.

### Epidemiological and clinical characteristics of mixed infections: Mechanisms and epidemiological implications

The finding that younger children were more susceptible to mixed infections (median age 3–4 years in mixed infection subgroups vs 8 years in the single infection group) is consistent with the 2025 China CDC surveillance report (24). This phenomenon may be due to the immature immune system of preschool children, which makes it difficult for them to resist infections by multiple pathogens (25). Notably, the mixed bacterial–viral infection subgroup had the highest premature delivery rate (8.77%), suggesting that preterm infants with inherent immune deficiencies may be at an even higher risk of mixed infections, which merits targeted clinical attention.

The higher incidence of mixed infections in autumn (28.22% vs 18.18% in the single infection group) is consistent with the epidemic characteristics of respiratory viruses. Autumn is the peak season for HRV, RSV, and other common viruses (26), and the overlap between viral epidemic seasons and MP infection cycles increases the probability of mixed infection. This seasonal pattern provides a basis for targeted prevention-strengthening respiratory protection and pathogen monitoring in autumn, especially for high-risk groups such as young children with a history of wheezing or pneumonia, potentially reducing the incidence of mixed infections.

An intriguing finding of our study was that children with mixed infections had a significantly higher proportion of severe pneumonia (30.44% vs 20.27%, $P < 0.001$) but paradoxically experienced a shorter median hospital stay (8 vs 9 days, $P < 0.001$) and lower rates of high fever (>39°C) with higher rates of low-to-moderate fever. This relationship between disease severity and both hospitalization duration and

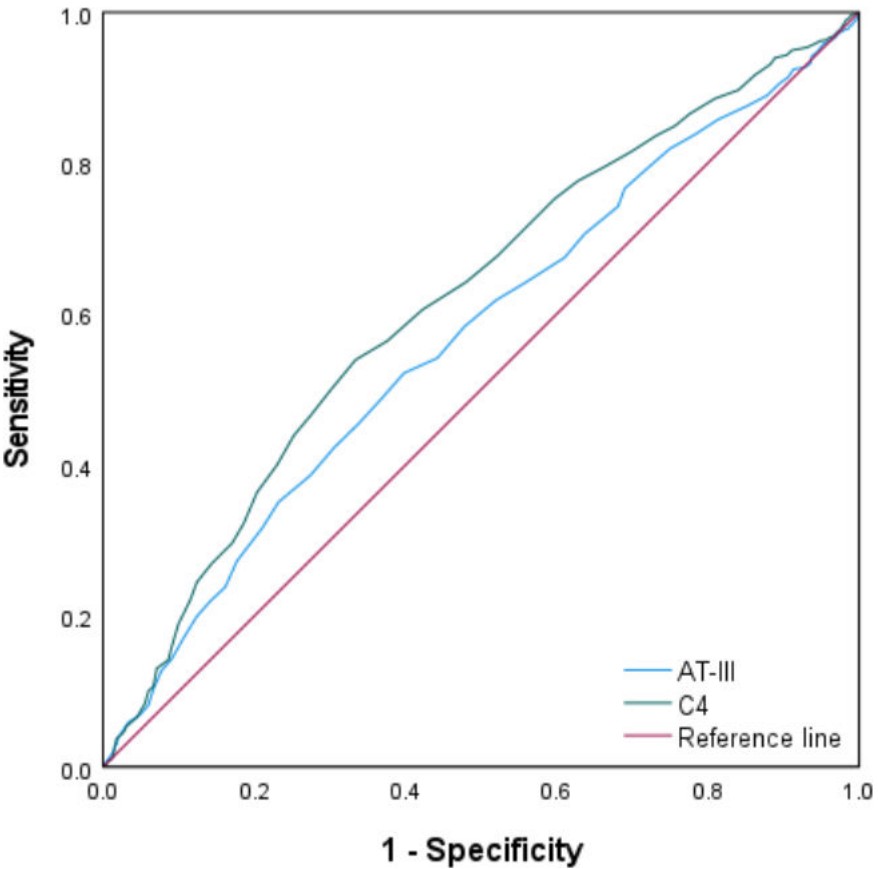

**FIG 1** ROC curves of AT-III and C4 for predicting mixed infection.

fever patterns challenges traditional perceptions. The shorter hospital stay despite greater severity might be explained by several factors: severe symptoms at onset likely prompted earlier health care seeking and timely admission; clinicians may have initiated more aggressive empirical therapy (e.g., combination antibiotics or antivirals) for suspected mixed infections, leading to faster pathogen clearance; and discharge practices might differ, with physicians more inclined to discharge mixed-infection patients early once the patient was clinically stable due to concerns about nosocomial infections or bed availability. Moreover, the lower fever intensity despite more severe pneumonia can be attributed to immune interference mechanisms proposed by Meyer et al. (27): the coexistence of viruses and *M. pneumoniae* may modulate the host systemic inflammatory response to viruses, which may inhibit excessive activation of proinflammatory pathways induced by *M. pneumoniae*, reducing the fever intensity while synergistically damaging the respiratory tract and increasing the disease severity. These findings highlight the limitations of using single indicators such as fever grade or length of stay to assess MPP severity, thus emphasizing the need for comprehensive evaluations combining clinical symptoms (e.g., rhinorrhea and wheezing) and laboratory indicators. Further research using standardized treatment protocols and discharge criteria is needed to clarify these relationships.

In terms of symptoms and signs, the mixed infection group had more prominent upper respiratory symptoms (nasal congestion, rhinorrhea) and wheezing, whereas the single infection group had more typical MPP features, such as high fever and pulmonary consolidation. This difference was related to the pathogenic characteristics of coinfecting pathogens: viruses such as HRV and HRSV primarily target the upper respiratory tract and induce airway hyperreactivity (28), whereas MP alone tends to cause interstitial lung inflammation and pulmonary consolidation (29). Additionally, the greater

**TABLE 6** Cut-off value of ROC curve for *M. pneumoniae* mixed-infection[a]

| Index | AUC (95% confidence interval） | Cut-off | Sensitivity (%) | Specificity (%) | P |
|---|---|---|---|---|---|
| AT-III | 0.570 (0.537–0.602) | 113.50 | 52.2 | 60.2 | **<0.001** |
| C4 | 0.617 (0.585–0.650) | 0.38 | 54.0 | 66.6 | **<0.001** |

[a]Bold column of *P*-values represents meaningful data.

proportion of previous pneumonia/wheezing history in mixed infection groups suggests that children with respiratory tract susceptibility may have impaired mucosal barrier function, increasing their risk of multipathogen invasion (30).

## Pathophysiological significance of laboratory indicator changes

### Immune function alterations

The levels of complement C3/C4 and immunoglobulin (IgA, IgG) were significantly reduced in the mixed infection group, demonstrating a particularly significant decrease in complement C4 in the mixed infection group. This finding is consistent with the conclusion proposed by Peng et al. that complement depletion is a key pathological link in mixed infections (31). As a core component of the classical pathway, complement C4 is essential for pathogen opsonization and clearance. When the host encounters multiple pathogens, excessive C4 consumption can lead to immune dysregulation and impaired pathogen elimination—a mechanism that not only explains the increased severity of mixed infections but also positions C4 as an active participant in disease progression rather than merely a passive biomarker. This mechanistic insight is supported by experimental evidence: Popescu et al. demonstrated that in the absence of C4, bacterial products cannot be efficiently opsonized, thereby compromising host defense against secondary bacterial infections and creating a permissive environment for mixed infections (32). Furthermore, genetic studies have shown that low levels of C4 resulting from complement gene polymorphisms are associated with an increased risk of viral coinfections such as HBV (33). However, the mechanism of complement consumption may be pathogen specific. Unlike *M. pneumoniae*, which primarily activates complement through immune complexes, *S. pneumoniae*, the predominant bacterial pathogen identified in our study, has a polysaccharide capsule that serves as a potent activator of the classical complement pathway, potentially leading to rapid consumption of early components such as C4 (34). Thus, the substantial C4 consumption observed in mixed infections may not merely represent a summation of inflammatory responses but rather reflect a more robust activation of the classical pathway driven by specific copathogens such as *S. pneumoniae*. This phenomenon provides a more precise mechanistic perspective for understanding why mixed infections lead to greater immune dysregulation and tissue damage. Collectively, these findings provide a coherent framework linking pathogen-specific complement activation, C4 consumption, immune dysfunction, and the pathogenesis of mixed infections. A decrease in IgA and IgG indicates impairment of both mucosal and systemic humoral immune function (35, 36), whereas a mild increase in IgM reflects an acute antibody response to newly invading pathogens.

The analysis of T lymphocyte subsets revealed that the proportions of CD3-CD (16 + 56)+% (NK cells) and CD3 + CD8+% (cytotoxic T cells) decreased in the mixed infection group. NK and CD8+ T cells are key effector cells for clearing viruses, and their decreased proportion may be related to the immune escape mechanism of dominant viral pathogens in mixed infections (37). For example, the human rhinovirus (HRV) with the highest detection rate in this study has been shown to encode a 3C protease that can cleave the host nuclear pore protein, interfere with the nuclear transport of interferon regulatory factor (IRF), and, thus, inhibit the production of type I interferon (38). This phenomenon may provide a new perspective on the immunological basis of the persistent presence of viral pathogens and the more severe clinical processes in mixed infections.

**TABLE 7** Seasonal distribution of viral pathogens[a]

| Virus | Number of detected pathogen (n, %) | Spring | Summer | Autumn | Winter | $\chi 2$ | P |
|---|---|---|---|---|---|---|---|
| HRV | 401 (45.46) | 127 | 111 | 113 | 50 | 4.359 | 0.225 |
| HRSV | 147 (16.67) | 70a | 21b | 34b, c | 22a, c | 33.372 | **<0.001** |
| HPIV | 87 (9.86) | 5a | 46b | 26b | 10b | 36.135 | **<0.001** |
| HCoV | 68 (7.71) | 9a | 50b | 7a | 2a | 65.178 | **<0.001** |
| HMPV | 42 (4.76) | 3a | 11a, b | 14b, c | 14c | 21.872 | **<0.001** |
| HBoV | 32 (3.63) | 3a | 8a, b | 17b | 4a, b | 14.687 | **0.002** |
| InfA H3N2 | 22 (2.49) | 6 | 9 | 6 | 1 | 2.246 | 0.523 |
| InfA H1N1 | 16 (1.81) | 15a | 0b | 1b | 0a, b | 31.263 | **<0.001** |
| HADV | 15 (1.70) | 5 | 3 | 2 | 5 | 5.966 | 0.113 |
| InfB | 8 (0.91) | 0a | 0a | 1a | 7b | 38.227 | **<0.001** |
| Ch | 5 (0.27) | 2 | 0 | 0 | 3 | 11.046 | **0.011** |
| Total | 882 (100) | 267 | 268 | 228 | 119 | –[b] | – |

[a]The subscript letters (a, b, c) for each number indicate no statistically significant difference if the same letter is present. Bold column of P-values represents meaningful data.
[b]"–" indicates that the chi-square value and P value have not been calculated for the corresponding indicator.

## Changes in coagulation markers

This study revealed significant increases and decreases in AT-III activity and Fib/DD-PLUS levels, respectively, in the mixed infection group. One study has shown that AT-III decreases during infection as a nutrition-dependent unstable protein in the post attack state and does not increase as an acute phase reactant (39). Studies have suggested that some of the beneficial effects of AT-III may stem from its direct anti-inflammatory effects rather than solely anticoagulation (40). As AT-III is an important endogenous anticoagulant, its elevation may constitute a compensatory mechanism aimed at maintaining microcirculation patency to combat inflammation-induced excessive coagulation activation (41). Studies have shown that in systemic inflammation (such as severe infection/sepsis), the coagulation system is widely activated, leading to the consumption of anticoagulant substances (including AT-III) (42). The compensatory feedback of the body may increase the production of AT-III through the upregulation of liver synthesis and other means, with the goal of restoring the anticoagulant procoagulant balance (43). The decreases in Fib and DD-PLUS suggest weak systemic fibrinolysis activation and a tendency toward hypercoagulability. This phenomenon may be related to the relatively mild systemic inflammatory response in mixed infections, as inflammation is the core driving force of the coagulation cascade (44).

## Changes in inflammatory markers

Notably, the levels of systemic inflammatory markers such as sCRP and SAA were lower in the mixed infection groups, a counterintuitive finding that has also been reported in other studies. This observation remains largely speculative and may be influenced by multiple factors. One potential explanation involves complex immune interactions between different pathogens. For instance, the CARDS toxin released by *M. pneumoniae* infection can induce abnormal and disordered host immune responses (45). When combined with *influenza virus*, viral proteins can negatively regulate excessive inflammatory responses by activating signaling pathways such as the PI3K/Akt pathway (46). Such overlapping immunomodulatory effects might theoretically attenuate the systemic acute-phase response, resulting in relatively lower sCRP and SAA levels at the time of detection. However, the timing of sample collection represents a critical alternative explanation. Inflammatory markers change dynamically during the course of infection, and slight differences in the sampling window may lead to substantial variations in

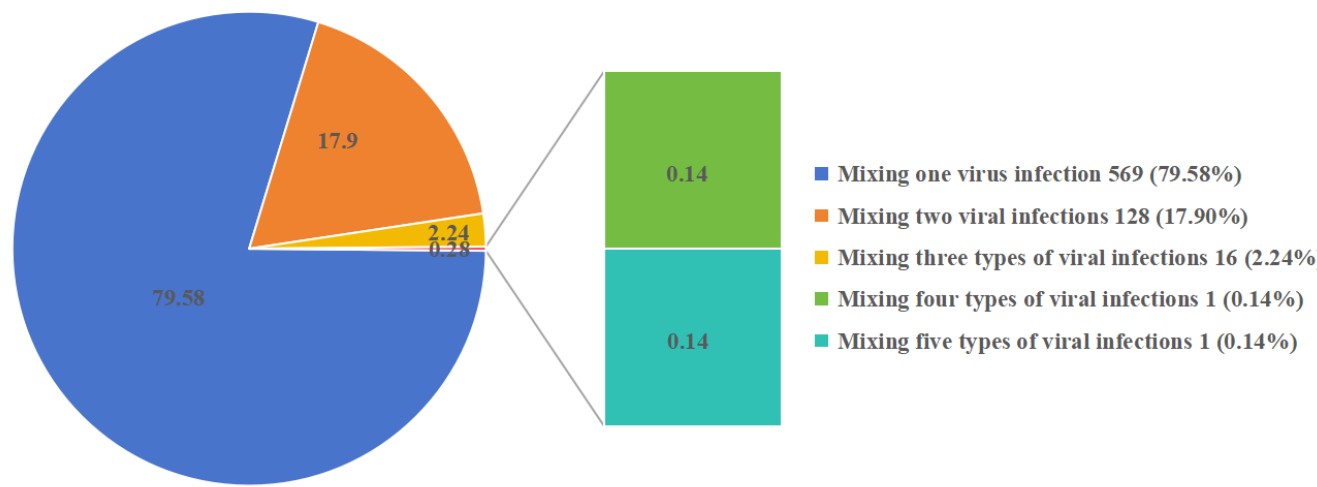

**FIG 2** Analysis of the proportion of infection types.

measured levels. Other unmeasured factors, including individual immune status and early antiviral or antibacterial treatment, may also contribute to this effect. Notably, these findings are not consistent across the literature. Tang et al. reported higher SAA levels in children with mixed MP infections (47). This discrepancy may be partly explained by differences in pathogen composition. Tang et al. included more proinflammatory bacteria, such as *H. influenzae*, whereas the present study was dominated by HRV and *S. pneumoniae*, which may elicit distinct inflammatory profiles. Collectively, these findings suggest that traditional inflammatory markers may not reliably reflect the severity of mixed infections. A combined panel including complement and coagulation indicators such as C4 and AT-III may provide a more comprehensive evaluation.

## Complications: Pathogenic mechanism differences

The incidence of gastrointestinal dysfunction and myocardial damage was significantly greater in the mixed infection group, which can be attributed to the synergistic effect of multiple pathogens. MP alone can cause extrapulmonary complications through immune mechanisms (e.g., molecular mimicry and immune complex deposition) or direct invasion (48). When complicated by bacterial/viral coinfection, MP and coinfecting pathogens (e.g., ADV and *S. pneumoniae*) synergistically activate both innate and adaptive immunity, trigger excessive inflammatory responses, and suppress regulatory T-cell function to exacerbate immune dysregulation, thereby further amplifying the systemic inflammatory load and damaging gastrointestinal and myocardial tissues (49). Notably, the severity of MP infection is positively correlated with the incidence of hypokalemia. In the single MP infection group, the incidence of hypokalemia is high, which is related mainly to the unique pathogenesis of MP, such as frequent vomiting, decreased appetite, and abnormal renal potassium excretion (50). In the mixed infection group, the incidence of hypokalemia was relatively low, which may have been due to other pathogens altering the electrolyte metabolism pathway or overlapping clinical manifestations with MP infection, thus masking the hypokalemia related to MP and leading to significant differences between the two groups.

## Risk factors and predictive value of mixed infections

Multivariate regression analysis revealed seven independent factors for mixed infections, among which age (OR = 0.609) and autumn onset (OR = 4.760) were practical epidemiological warning indicators. Each 1-year increase in age reduced the risk by ~40%, and autumn was associated with a nearly fivefold increase in risk. Rhinorrhea (OR = 7.917) is a strong positive predictor, which is highly consistent with the involvement of respiratory

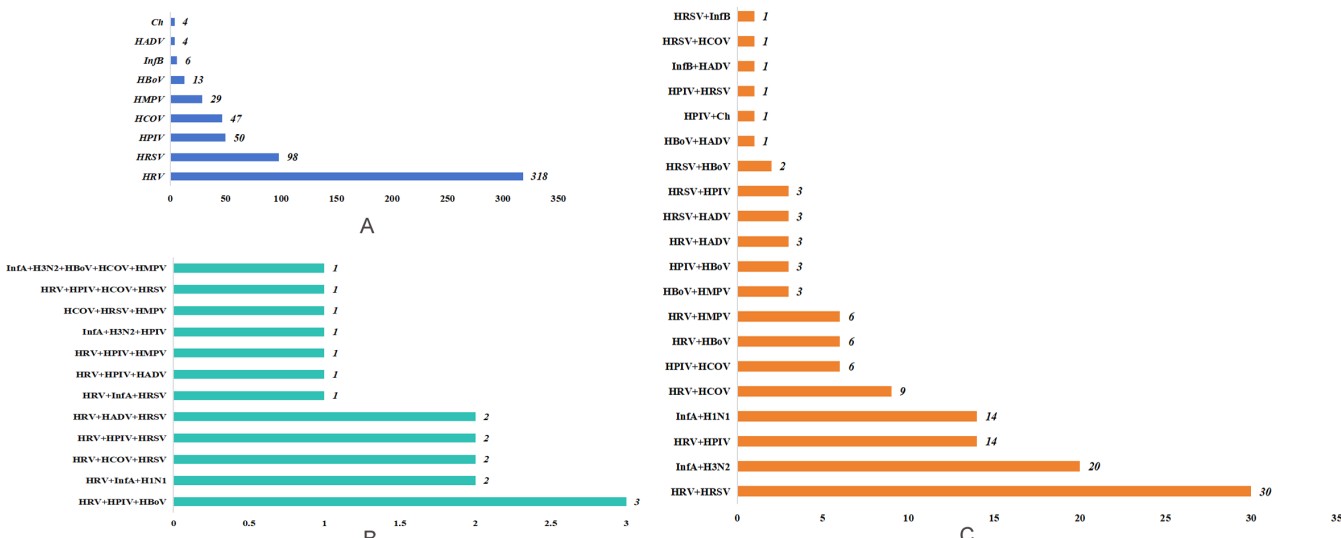

**FIG 3** Detailed analysis of mixed infection combinations (A) mixing one virus infection; (B) mixing three or more viral infections; (C) mixing two viral infections).

viruses in mixed infections, whereas vomiting (OR = 0.155) is a protective factor, possibly because it is more specific to single MP infections. The AUC values of AT-III (0.570) and C4 (0.617) suggested a relatively low predictive efficacy and unsatisfactory accuracy when they were used as individual indicators. Although these indicators were independently associated with mixed infections, their predictive value as single markers was limited and only provided certain auxiliary reference significance. Therefore, to improve the predictive performance, it is recommended to combine these two laboratory indicators with epidemiological and clinical factors, such as age, season, vomiting, and rhinorrhea, to construct a multidimensional predictive model. This approach is similar to the pediatric infectious disease prediction system established by Liu et al. (51) which integrates nine laboratory and clinical indicators to enable rapid bedside assessment. Such a comprehensive model could help clinicians identify high-risk children at an early stage, particularly those with bacterial or viral mixed infections, and thus avoid delayed diagnosis and treatment. Future studies are warranted to develop a multivariate predictive model incorporating clinical variables and laboratory parameters to achieve more robust and reliable predictions, providing a clearer direction for subsequent related research.

## Pathogen spectrum characteristics and clinical implications

### Viral spectrum

In this study, MP + single-virus infection (79.58%) was the main mode of infection. The detection rate of HRV was the highest (45.46%), with a stable distribution throughout the year, which is consistent with reports identifying HRV as the main pathogen in young children (52). This was followed by RSV, which accounted for 16.67% and showed seasonal characteristics peaking in spring (70/147). Following RSV was HPIV (9.86%), which was concentrated in summer (46/87). These results indicated that the duration of the concentrated distribution of different viruses varied, providing a basis for clinical seasonal prevention and control. The overall trend is consistent with the monitoring data of the Global Respiratory Virus Epidemiology Map released by the World Health Organization in 2023 (53).

However, this lineage characteristic has obvious regional characteristics. A multicenter study in North America indicated that the most common viruses associated with mixed infections in hospitalized children with MPP were RSV and HRV (54). A study conducted in *Russia* reported that HPIV and SARS-CoV-2 were both common pathogens

**TABLE 8** Annual distribution of bacterial pathogens

| Pathogens | 2021 (*n*, %) | 2022 (*n*, %) | 2023 (*n*, %) | Number of detected strains (*n*, %) |
|---|---|---|---|---|
| Gram-positive bacteria | 103 (28.69) | 101 (28.13) | 155 (43.18) | 359 (75.42) |
| *Streptococcus pneumoniae* | 92 (30.07) | 77 (25.16) | 137 (44.77) | 306 (64.29) |
| *Staphylococcus aureus* | 10 (20.41) | 23 (46.94) | 16 (32.65) | 49 (10.29) |
| *Streptococcus dysgalactiae* | 1 (50) | 1 (50) | 0 | 2 (0.42) |
| *Streptococcus pyogenes* | 0 | 0 | 1 (100) | 1 (0.21) |
| *Other Staphylococcus species* | 0 | 0 | 1 (100) | 1 (0.21) |
| Gram-negative bacteria | 22 (18.80) | 56 (47.86) | 39 (33.33) | 117 (24.58) |
| *Haemophilus influenzae* | 19 (21.11) | 40 (44.44) | 31 (34.44) | 90 (18.91) |
| *Pseudomonas aeruginosa* | 3 (50) | 3 (50) | 0 | 6 (1.26) |
| *Klebsiella species* | 0 | 2 (33.33) | 4 (66.67) | 6 (1.26) |
| *Moraxella catarrhalis* | 0 | 3 (60) | 2 (40) | 5 (1.05) |
| *Escherichia coli* | 0 | 2 (100) | 0 | 2 (0.42) |
| *Acinetobacter species* | 0 | 2 (100) | 0 | 2 (0.42) |
| *Stenotrophomonas maltophilia* | 0 | 0 | 2 (100) | 2 (0.42) |
| *Enterobacter kobei* | 0 | 1 (100) | 0 | 1 (0.21) |
| *Uralobacter terricola* | 0 | 1 (100) | 0 | 1 (0.21) |
| *Enterobacter cloacae* | 0 | 1 (100) | 0 | 1 (0.21) |
| *Serratia marcescens* | 0 | 1 (100) | 0 | 1 (0.21) |
| Total | 125 (26.26) | 157 (32.98) | 194 (40.76) | 476 (100) |

in these cases (55). A study from Liaoning Province, China, demonstrated that ADV was the most prevalent pathogen in MPP patients with mixed infections (56). Findings from Beijing, China, revealed that the most common bacterial copathogen was *S. pneumoniae*, while the predominant viral copathogens were influenza virus and HPIV (57). In a study focused on acute respiratory tract infections in children in Beijing during spring 2023, Sun et al. reported that the primary causative viruses during this period were RSV and *influenza virus* (58). This difference might be due to the following factors. (i) Climate and environmental factors: In the East China region where this study was located, the warm and humid climate might be more conducive to the year-round transmission of HRV. (ii) Population immunity background: Children from different regions have different exposure histories and population immunity levels, which affect the dominant epidemic strains. (iii) Age and population composition: This study was focused on children with MPP whose respiratory microenvironments might be more susceptible to specific viruses such as HRV. These comparisons suggest that clinical experience and prevention and control strategies cannot be simply applied across regions and that establishing a localized and dynamic pathogen monitoring network is crucial.

The infection mode of MP + dual virus accounted for 17.90% of the infections. Because HRV, HRV, and HPIV were the three most frequently detected viruses, HRV + HRSV (30 cases) and HRV + HPIV (14 cases) were the most common combinations of the two viruses in the present study. This finding is consistent with the findings of Jain et al. that HRV + RSV mixed infection is the main cause of MPP in children (54). Triple or more viral coinfections are rare, but caution is needed, as they may trigger cytokine storms in immunocompromised children, leading to serious illnesses.

### Bacterial spectrum

Compared with gram-negative bacteria, gram-positive bacteria were the absolute dominant flora (75.42%) and were significantly more abundant (24.58%), exhibiting a close relationship to the physiological characteristics of the children's respiratory mucosa and bacterial colonization patterns. *S. pneumoniae* was the dominant bacterium in MP

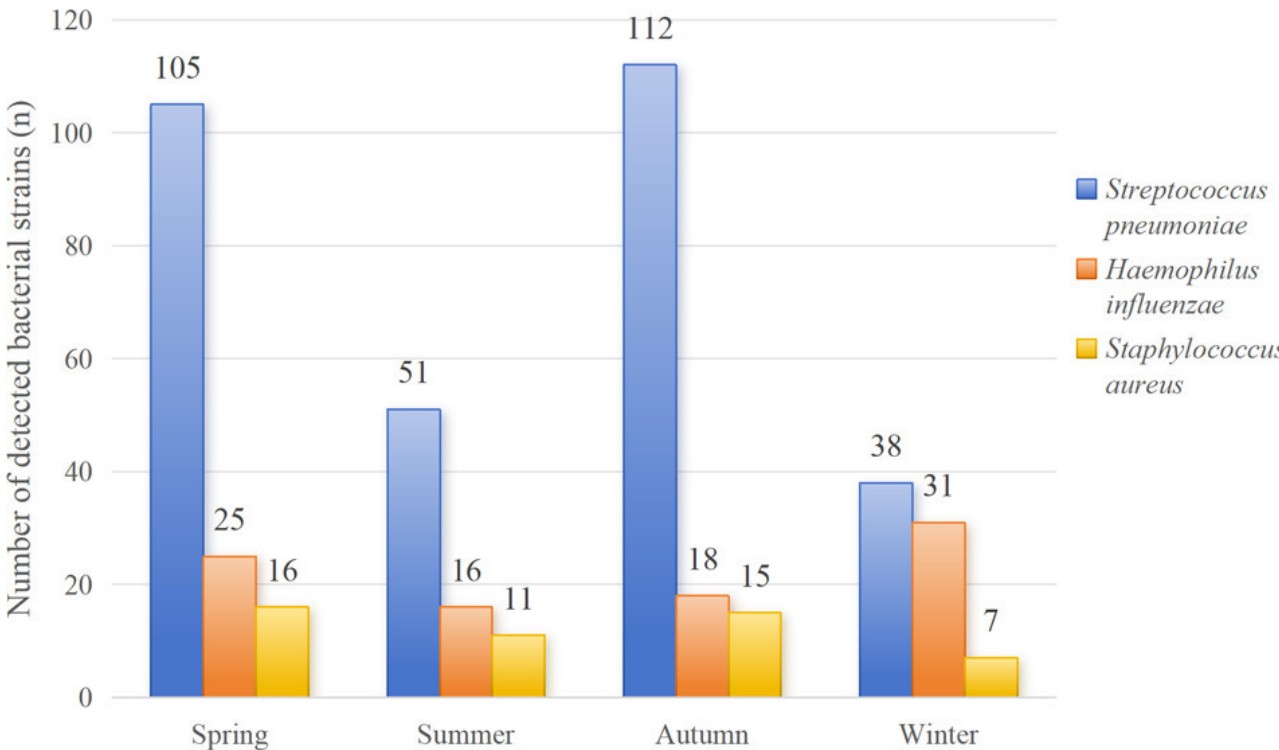

**FIG 4** Seasonal analysis of common bacteria.

mixed infections, accounting for 64.29% of all the infections. This finding is consistent with the conclusions of multiple domestic studies, establishing its position as the main "bacterial companion" of MP (59, 60). Notably, the detection rate of *S. pneumoniae* will reach a significant peak in 2023 (44.77%), which is consistent with the "Report on the Epidemic Trends of Respiratory Pathogens in Children" released by the Chinese CDC in 2025 (covering 31 provinces across the country) (61).

In 2023, the national percentage of children infected with *S. pneumoniae* was expected to increase 37.6% compared with that in 2021. The core driving factor may be related to the "immune debt" phenomenon after the liberalization of COVID-19 prevention and control measures, leading to the rebound and transformation of the epidemic mode of common respiratory pathogens (62). Research published in 2025 confirmed that after relaxation of prevention and control measures for COVID-19, the mixed infection rate of infants under 5 years old was the highest, reaching 52.3%, of which the mixed infection rate of *S. pneumoniae* accounted for 38.7% (63). Since the introduction of the pneumococcal vaccine (PCV13) into China's immunization program in 2022, the infection rate of some serotypes has declined, but recent studies have reported that the resurgence of invasive pneumococcal diseases in children after COVID-19 is driven mainly by serotype 3, which is related to more serious diseases, PICU admission, and fatal outcomes (64). This observation emphasizes the necessity of improving vaccine formulations, continuous monitoring, and timely adjustment of immunization strategies.

Although the overall proportion of *H. influenzae,* as the main gram-negative bacterium, was lower than that of *S. pneumoniae* (18.91% < 64.29%), it is a common colonizer of the upper respiratory tract of children, similar to *S. pneumoniae*, and can take advantage of the weakened respiratory defenses caused by MP infection to cause synergistic infection. According to reports, mixed infection with *H. influenzae* and MP is associated with more severe clinical manifestations, especially airway hypersecretion and wheezing. Compared with the single *H. influenzae* infection group (21.5%) and the single MP infection group (28.7%), the mixed infection group showed a significant increase in airway mucus secretion and a significantly higher incidence of wheezing (42.3%) (65).

The annual distribution of *S. aureus* tended to increase but then decrease, peaking in 2022 (46.94%) and declining to 32.65% in 2023. This fluctuation warrants high vigilance, as *S. aureus*, particularly community-associated methicillin-resistant *S. aureus* infection, is often associated with severe complications such as severe pneumonia and necrotizing pneumonia (66). Its annual fluctuation highlights the need for continuous monitoring to determine whether these changes represent a new epidemic trend. In summary, this study reveals that the bacterial spectrum is dominated by *S. pneumoniae* and *H. influenzae*, as well as their fluctuations over time (such as a significant increase in the detection rate of *S. pneumoniae* in 2023). This pattern highlights the urgency of establishing a regional, long-term, and dynamic monitoring network for respiratory bacterial pathogens in children, suggesting that when children with highly suspected mixed infections are empirically treated, narrow spectrum or targeted antibiotics that can cover these common pathogens should be prioritized.

In this study, the seasonal distributions of major bacterial pathogens were analyzed, revealing significant seasonal differences in the prevalence of *S. pneumoniae* and *H. influenzae*. *S. pneumoniae* was the dominant pathogen in spring (105 strains), summer (51 strains), and autumn (112 strains), with a significantly greater number of detections than *H. influenzae* and *S. aureus* during the same periods, peaking in autumn (112 strains). In winter, however, the number of *S. pneumoniae* detections decreased to the lowest level of the year (38 strains), and compared with that of *H. influenzae* (31 strains), the dominant position of *S. pneumoniae* was markedly weakened with a greatly narrowed gap. In contrast, *H. influenzae* showed a distinct winter peak, with the highest number of detections in winter (31 strains), which was significantly greater than that in spring (25 strains), summer (16 strains), and autumn (18 strains), making it the closest major pathogen to *S. pneumoniae* in winter. In conclusion, *S. pneumoniae* was the absolute dominant pathogen in spring, summer, and autumn, whereas *H. influenzae* was more active in winter, demonstrating a seasonal mismatch in its epidemic peaks.

## Strengths and limitations of the study

The strengths of this study include the following: (i) the large sample size (1,428 cases) to ensure the statistical power; (ii) the use of comprehensive pathogen detection (13-plex nucleic acid testing + bacterial culture) to clarify the full spectrum of mixed infections; (iii) the systematic analysis of immune, coagulation, and inflammatory indicators to reveal the pathophysiological mechanism of mixed infections; and (iv) the identification of independent risk factors and the construction of a potential prediction model with direct clinical application value.

However, this study is a retrospective observational study aimed primarily at describing epidemiological characteristics, identifying risk factors, and exploring biomarkers. Although we proposed a predictive model based on risk factors and biomarkers, the data from this study cannot provide direct, evidence-based decision-making guidance for specific, stratified antimicrobial drug selection schemes. Translating the pathogen spectrum characteristics and risk models discovered in this study into actionable, personalized guidelines for antimicrobial drug management requires validation in future prospective intervention studies. In addition, this was a single-center study conducted in Suzhou, China, which limits the generalizability of our findings. The epidemiological characteristics, including the autumn peak and predominant pathogens (e.g., HRV and *S. pneumoniae*), were influenced by region, climate, and population. These results are applicable only to regions with similar climates, such as eastern and southern China, and cannot be directly extrapolated to other regions of China with distinct climates or other countries. Thus, caution should be exercised when interpreting the external validity of our findings.

## Clinical implications and future directions

This study provides important clinical insights as follows: (i) for young children (especially those <4 years old) with MPP who present in autumn with rhinorrhea and prolonged cough, mixed infection should be highly suspected; (ii) combined detection of complement C4 and AT-III can assist in early identification of mixed infections, reducing the risk of severe pneumonia; (iii) targeted prevention measures should be implemented according to pathogen seasonal characteristics (e.g., HRV in autumn and HRSV in spring); and (iv) rational use of antimicrobial agents should be guided by pathogen spectrum data, avoiding the overuse of broad-spectrum antibiotics.

## ACKNOWLEDGMENTS

We thank the staff from the Department of Clinical Laboratory, Children's Hospital of Soochow University, who took part in the study.

This research was supported by the Special Foundation for National Science and Technology Basic Research Program of China (2019FY101200), the High-level Innovative and Entrepreneurial Talents Introduction Program of Jiangsu Province (2020-30191), Medical Research Project of Jiangsu Commission of Health (M2020027), National Tutorial System Project of Suzhou Health Young Backbone Talents (Qngg2022011), Key Research Program Cultivation Project of Children's Hospital of Soochow University (2023ZDPY02), Suzhou Municipal Health Commission (KJXW2023025), Gusu Health Talent Program of Suzhou (2024-094), the Science and Technology Program of Suzhou (SYW2024103), Government Scholarship for Studying Abroad of Jiangsu Province (2024-087), Scientific Research Project of Jiangsu Provincial Maternal and Child Health Care Association (FYX202505), Science and Technology Project of Suzhou Municipal Health Commission (MSXM2025019), Guided Project of Jiangsu Provincial Natural Science Foundation (Z2025037).

X.Z. and C.W. conceived the study and designed the experiments. Y.L., X.S., and X.Z. provided financial support. C.W., S.H., and L.Y. collected the data, Y.G. and L.Y. analyzed the data, X.Z. and J.X. interpreted the results. X.Z. and C.W. drafted the manuscript, and all authors critically revised the manuscript for intellectual content, and read and approved the final manuscript.

The views, opinions, assumptions, or any other information set out in this article are solely those of the authors and should not be attributed to the funders or any other person connected with the funders.

## AUTHOR AFFILIATIONS

[1]Department of Clinical Laboratory, Children's Hospital of Soochow University, Suzhou, China
[2]Institute of Pediatric Research, Children's Hospital of Soochow University, Suzhou, China

## AUTHOR ORCIDs

Shenghao Hua  http://orcid.org/0009-0000-4422-9139
Xuejun Shao  http://orcid.org/0000-0002-4853-2634
Jun Xu  http://orcid.org/0009-0009-0644-6744
Xin Zhang  http://orcid.org/0009-0008-3567-1791

## FUNDING

| Funder | Grant(s) | Author(s) |
| --- | --- | --- |
| Special fundation for national science and technology basic research program of china | 2019FY101200 | Xuejun Shao |
| Scientific Research Project of Jiangsu Provincial Maternal and Child Health Care Association | FYX202505 | Yang Li |

| Funder | Grant(s) | Author(s) |
|---|---|---|
| Science and Technology Project of Suzhou Municipal Health Commission | MSXM2025019 | Yang Li |
| Guided Project of Jiangsu Provincial Natural Science Foundation | Z2025037 | Xin Zhang |
| High Level Innovation and Entrepreneurial Research Team Program in Jiangsu (Expert Teams of Program of Innovation and Entrepreneurship of Jiangsu Province) | 2020-30191 | Yang Li |
| Government scholarship for studying abroad of jiangsu province | 2024-087 | Yang Li |
| Medical research project of jiangsu commission of health | M2020027 | Yang Li |
| National tutorial system project of suzhou health young backbone talents | Qngg2022011 | Yang Li |
| Gusu health talent program of suzhou | 2024-094 | Yang Li |
| Science and Technology Program of Suzhou | SYW2024103 | Yang Li |
| Key research program cultivation project of children's hospital of soochow university | 2023ZDPY02 | Yang Li |
| Suzhou Municipal Health Commission | KJXW2023025 | Xin Zhang |

## AUTHOR CONTRIBUTIONS

Cheng Wang, Conceptualization, Data curation, Writing – original draft | Shenghao Hua, Data curation, Project administration | Yang Li, Data curation, Funding acquisition | Yuanyuan Gao, Formal analysis, Project administration | Lunjing Yan, Formal analysis, Project administration | Xuejun Shao, Formal analysis, Funding acquisition, Project administration | Jun Xu, Formal analysis, Project administration, Writing – review and editing | Xin Zhang, Conceptualization, Funding acquisition, Project administration, Writing – review and editing

## DATA AVAILABILITY

The data sets used and/or analyzed during the current study are available from the corresponding author on reasonable request.

## ETHICS APPROVAL

The project is a retrospective study of data and only includes specific data. It was reviewed and approved by the Medical Ethics Committee of the Children's Hospital of Soochow University (Ethics batch number: 2025CS239).

## ADDITIONAL FILES

The following material is available online.

### Supplemental Material

**Supplemental Material (Spectrum03884-25-s0001.docx).** Tables S1 to S6.

### Open Peer Review

**PEER REVIEW HISTORY (review-history.pdf).** An accounting of the reviewer comments and feedback.

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
