## [Reviewer comments · Microbiology Spectrum]

Microbiology Spectrum

Clinical Characteristics, Risk Factors and Pathogen Spectrum of Mixed Infections in Children with *Mycoplasma pneumoniae* Pneumonia: A Retrospective Cohort Study of 1,428 Cases from a Children's Medical Centre in East China

Cheng Wang, Sheng-Hao Hua, Yang Li, Yuanyuan Gao, Lunjing Yan, Xuejun Shao, Jun Xu, and Xin Zhang

Corresponding Author(s): Xin Zhang, Children's Hospital of Soochow University

Review Timeline:

Submission Date:	December 2, 2025
Editorial Decision:	December 20, 2025
Revision Received:	January 17, 2026
Editorial Decision:	February 27, 2026
Revision Received:	March 31, 2026
Accepted:	April 2, 2026

Editor: Ping Ren

Reviewer(s): Disclosure of reviewer identity is with reference to reviewer comments included in decision letter(s). The following individuals involved in review of your submission have agreed to reveal their identity: Xinxing Zhang (Reviewer #1); Yue Tao (Reviewer #2); Qing Cao (Reviewer #3)

Transaction Report:

DOI: <https://doi.org/10.1128/spectrum.03884-25>

Re: Spectrum03884-25 (**Clinical Characteristics, Risk Factors and Pathogen Spectrum of Mixed Infections in Children with *Mycoplasma pneumoniae* Pneumonia: A Retrospective Cohort Study of 1,428 Cases from a Children's Medical Centre in East China**)

Dear Ms. Xin Zhang:

I have received the reviews of your manuscript and regret to inform you that we will not be able to publish it in *Microbiology Spectrum*. Your submission was read by reviewers with expertise in the area addressed in your study and it was the consensus view of these reviewers that your paper did not meet the standards necessary for publication.

I am sorry to convey a negative decision on this occasion, but I hope that the enclosed reviews are useful. Please note, rejections from *Microbiology Spectrum* are final and your manuscript will not be considered by other ASM journals. We wish you well in publishing this report in another journal and hope that you will consider *Spectrum* in the future.

Sincerely,
Ping Ren
Editor, *Microbiology Spectrum*

Reviewer #1 (Comments for the Author):

This study provides a comprehensive retrospective analysis of mixed infections in children with *Mycoplasma pneumoniae* pneumonia (MPP). The research is well designed, with a large sample size and clear conclusions that hold significant clinical relevance. However, several sections require revision to improve clarity, depth, and scientific rigor.

1. Introduction

Research needs to strengthen the targeting of research gaps. Currently, only "insufficient research on pathogen spectrum and risk factors" is mentioned. Specific comparisons should be made between the pathogen differences of MPP mixed infections in different regions at home and abroad (such as North China, South China, and Europe and America), highlighting the unique epidemiological background of children in East China and clarifying the filling value of this study.

2. Methods

The "13 plex respiratory pathogen nucleic acid testing" should list all 13 pathogens in the Methods section rather than only in the results tables, to ensure transparency and reproducibility. Classification of influenza viruses: "H3N2" and "H1N1" are listed separately in Table 5 but should be clearly categorized under "Influenza virus (InfA/InfB)" in the Methods, with a note that they represent subtypes of InfA, to avoid confusion.

3. Results

3.1 In Table 1, cough duration: Both groups show a median of 6 days, but the interquartile ranges differ (4-9 days for mixed infection vs. 4-7 days for single infection). Please provide the non parametric test statistic (e.g., Z value) in addition to the P value.

3.2 Bacterial spectrum seasonality: Presenting the annual/seasonal distribution only in a table limits readability. Adding a visual display (e.g., a heatmap, line chart, or stacked bar chart) would help readers grasp trends more intuitively.

3.3 Complications: There is an apparent calculation error for sepsis in the mixed infection group: $9/900 = 1.00\%$, not 10%. Please thoroughly verify all percentages, counts, and corresponding P values throughout the manuscript.

4. Discussion (The discussion is well written and data driven, but it tends to reiterate results rather than explore underlying mechanisms.)

4.1 Mechanism and literature comparison: Strengthen the comparison with other studies-not only highlighting consistencies but also discussing discrepancies, deepen the mechanistic explanation.

4.2 Regional comparison of pathogen spectrum: Systematically compare your pathogen profile with those reported in other regional studies (both domestic and international) to illustrate whether your findings are region specific or more universal.

4.3 Move from description to interpretation: Instead of simply stating "we found lower C4 in the mixed infection group," explain why this might occur and what it implies. For example: "The lower complement C4 level in mixed infections suggests excessive consumption of the complement pathway when the host confronts multiple pathogens, which may contribute to immune dysregulation and more severe disease. This aligns with the findings of Andrejeva et al., indicating that C4 is not merely a biomarker but potentially an active player in disease progression."

4.4 Improve structure: The discussion currently appears as a single, dense block of text. Adding subheadings (e.g., Summary of

key findings, Clinical and epidemiological profile, Immunological and coagulation characteristics, Pathogen spectrum and synergy, Clinical implications and future directions) would greatly enhance readability.

5. Formatting and Stylistic Issues

5.1 Abbreviations: Ensure that all abbreviations are spelled out at first use (e.g., "antithrombin III (AT III)").

5.2 Table consistency: Unify the presentation of data in tables (use "n (%)" throughout) and ensure a space between numbers and units (e.g., "8 days" not "8 days").

5.3 References: Check that all references include volume, issue, and page numbers where required, and correct any incomplete entries.

Recommendation :

Minor revision is recommended. Addressing these points will significantly strengthen the manuscript's scientific rigor, clarity, and impact.

Reviewer #3 (Comments for the Author):

Clinical Characteristics, Risk Factors and Pathogen Spectrum of Mixed Infections in Children with *Mycoplasma pneumoniae* Pneumonia: A Retrospective Cohort Study of 1,428 Cases from a Children's Medical Centre in East China, Single-center (2021-2023).

Major Concerns and Areas for Improvement

1. Language, Grammar, and Writing Quality [Needs Major Revision]

The manuscript contains numerous instances of unclear, awkwardly phrased, or grammatically incorrect English. Many sentences are excessively long, lack logical flow, or employ imprecise terminology. These issues significantly compromise the clarity, professionalism, and readability of the study.

Recommendations:

Professional English language editing is essential. Consider utilizing services from native-speaking scientific editors (e.g., Elsevier Language Services, Editage, or Nature Research Editing Service).

Ensure consistent terminology throughout the manuscript (e.g., standardize the use of "mixed infection" versus "co-infection"). Simplify overly complex or redundant sentences to enhance the academic tone and overall clarity.

2. Methods Section: Lack of Critical Details

While the methodological approach is generally sound, the omission of several key details affects the study's reproducibility and the assessment of its validity.

2.1 Pathogen Detection Assays:

For PCR-based tests (e.g., the 13-plex Respiratory Pathogen Panel), provide information on the specific commercial kit, manufacturer, primer/probe sequences, detection limits, and validated targets.

For *Mycoplasma pneumoniae* IgM (MP-IgM) testing, specify the assay's sensitivity and specificity, as well as the reference values used to define positivity.

For bacterial culture, detail the colony count thresholds considered positive, quality control procedures for media, and the standards for microbial identification.

2.2 Laboratory Biomarkers:

While numerous biomarkers are listed (e.g., C3, C4, IgA, IgG, AT-III, sCRP, SAA), the methodologies, specific assay kits, reference ranges, and units of measurement (beyond the few examples in table footnotes) are not provided.

Table 2 presents several biomarkers without explaining the clinical significance of their abnormal values.

2.3 Data Quality and Handling of Missing Data:

Describe how missing data were addressed, if present.

Specify whether data extraction was performed in duplicate or by trained personnel to ensure accuracy.

Consider mentioning whether blinding was employed during data analysis, as this enhances transparency even in retrospective studies.

3. Statistical Analysis: Limited Interpretation and Reporting

Although basic statistical methods (e.g., Mann-Whitney U test, chi-square test, logistic regression) are appropriately applied, several important elements are underreported or insufficiently interpreted.

3.1 Multivariate Regression:

Report measures of model fit (e.g., the Hosmer-Lemeshow test) and diagnostics for multicollinearity (e.g., Variance Inflation Factor, VIF).

Discuss potential confounders that were evaluated but not included in the final model.

3.2 ROC Analysis:

For the ROC curves presented for AT-III and C4, provide not only the Area Under the Curve (AUC) values and optimal cut-offs but also the corresponding sensitivity, specificity, positive/negative predictive values, and likelihood ratios, which are crucial for

assessing clinical utility.

Discuss the clinical rationale or implications of the chosen cut-off points.

3.3 Subgroup Comparisons:

Conduct and report direct comparisons between different types of mixed infections (e.g., bacterial-only vs. viral-only vs. bacterial+viral).

Consider performing stratified analyses by age groups (e.g., <3 years, 3-6 years, >6 years) or by seasonal subsets to explore potential effect modifiers.

4. Results: Presentation and Interpretation

4.1 Strengths:

Clear tabular presentation of demographic data, clinical features, and laboratory results.

Appropriate identification of key differences between the single and mixed infection groups.

4.2 Areas for Improvement:

4.2.1 Some tables (e.g., Tables 1 and 3) are densely packed with numbers, percentages, and p-values, making them difficult to interpret quickly. Consider streamlining or using supplementary tables for detailed data.

4.2.2 Several clinically relevant outcomes are not reported in detail, such as: ICU admission rates; Need for and duration of oxygen support; Requirement for mechanical ventilation; Use of adjunctive therapies like corticosteroids or intravenous immunoglobulin (IVIG), which are common in severe *M. pneumoniae* pneumonia.

4.2.3 Some findings lack deeper interpretation. For instance:

Why might gastrointestinal dysfunction and myocardial injury be more common in mixed infections? What could explain the higher frequency of hypokalemia in the single infection group? What is the potential mechanistic or immunological basis for the observed differences in C4 and AT-III levels?

5. Discussion: Needs More Depth and Consideration of Limitations

The Discussion section is somewhat repetitive of the Results, offering limited new synthesis or insight. There is also insufficient discussion on how the study's findings could inform clinical practice or guide antimicrobial stewardship.

Title: Clinical Characteristics, Risk Factors and Pathogen Spectrum of Mixed Infections in Children with *Mycoplasma pneumoniae* Pneumonia: A Retrospective Cohort Study of 1,428 Cases from a Children's Medical Centre in East China

General Assessment:

This manuscript presents a well-conducted, large-scale retrospective study investigating the important clinical problem of mixed infections in pediatric *Mycoplasma pneumoniae* pneumonia (MPP). The study is methodologically sound, the statistical analysis appears rigorous, and the findings regarding the high prevalence (63.03%), distinct clinical and laboratory profiles, and independent risk factors for mixed infections are significant and clinically relevant.

Major Comments:

1. **Apparent Paradox in Clinical Outcomes:** The data show that the mixed infection group had a significantly **shorter hospital stay** (8 vs. 9 days, $p < 0.001$) yet a **higher incidence of severe pneumonia** (30.44% vs. 20.27%, $p < 0.001$). This seemingly contradictory finding is not adequately discussed. Please provide a plausible explanation in the Discussion section. Potential reasons could include earlier onset of severe symptoms leading to prompt hospitalization, differences in treatment protocols or response for mixed infections, or variations in discharge criteria. Clarifying this point is crucial for a coherent interpretation of the disease burden.
2. **Pathophysiological Rationale for Key Biomarkers:** The identification of decreased complement C4 and elevated antithrombin III (AT-III) as independent protective and risk factors, respectively, is interesting. However, the discussion on their biological plausibility in the context of mixed infections is relatively brief. Please expand the discussion to more explicitly link these findings to the underlying immunology and coagulation dynamics. For C4, elaborate on how complement consumption or dysregulation might predispose to or result from co-infection. For AT-III, discuss its role as an acute-phase reactant and its

potential elevation as a compensatory anti-inflammatory or anticoagulant response in mixed infections, referencing relevant literature.

3. **Limitations and Generalizability:** The single-center design from Suzhou, East China, is appropriately noted as a limitation. To better contextualize the findings, please briefly discuss the potential implications of this limitation. For instance, are the epidemiological patterns (e.g., autumn peak, dominant pathogens like HRV and *S. pneumoniae*) likely representative of other regions in China or other countries with similar climates/seasonality? Acknowledging this will help readers gauge the external validity of the results.

Minor Comments:

4. **Clarity on Bacterial Seasonality:** The viral pathogen spectrum is nicely analyzed for seasonal variation (Table 5). Please consider adding a sentence or two in the Results or Discussion regarding whether any **seasonal trends were observed for the main bacterial pathogens** (*S. pneumoniae*, *H. influenzae*), as this would provide a more complete epidemiological picture.
5. **Prediction Model Suggestion:** The conclusion recommends developing a prediction model. Since the ROC analysis for C4 and AT-III shows modest AUCs (0.617 and 0.570), it would be helpful to state more explicitly that these two biomarkers alone have limited discriminatory power and that a **future multivariable model integrating clinical factors (age, season, symptoms)** is needed for robust prediction. This sets a clearer direction for subsequent research.

Clinical Characteristics, Risk Factors and Pathogen Spectrum of Mixed Infections in Children with *Mycoplasma pneumoniae* Pneumonia: A Retrospective Cohort Study of 1,428 Cases from a Children's Medical Centre in East China, Single-center (2021-2023).

Major Concerns and Areas for Improvement

1. Language, Grammar, and Writing Quality [Needs Major Revision]

The manuscript contains numerous instances of unclear, awkwardly phrased, or grammatically incorrect English. Many sentences are excessively long, lack logical flow, or employ imprecise terminology. These issues significantly compromise the clarity, professionalism, and readability of the study.

Recommendations:

Professional English language editing is essential. Consider utilizing services from native-speaking scientific editors (e.g., Elsevier Language Services, Editage, or Nature Research Editing Service).

Ensure consistent terminology throughout the manuscript (e.g., standardize the use of "mixed infection" versus "co-infection").

Simplify overly complex or redundant sentences to enhance the academic tone and overall clarity.

2. Methods Section: Lack of Critical Details

While the methodological approach is generally sound, the omission of several key details affects the study's reproducibility and the assessment of its validity.

2.1 Pathogen Detection Assays:

For PCR-based tests (e.g., the 13-plex Respiratory Pathogen Panel), provide information on the specific commercial kit, manufacturer, primer/probe sequences, detection limits, and validated targets.

For *Mycoplasma pneumoniae* IgM (MP-IgM) testing, specify the assay's sensitivity and specificity, as well as the reference values used to define positivity.

For bacterial culture, detail the colony count thresholds considered positive, quality control procedures for media, and the standards for microbial identification.

2.2 Laboratory Biomarkers:

While numerous biomarkers are listed (e.g., C3, C4, IgA, IgG, AT-III, sCRP, SAA), the methodologies, specific assay kits, reference ranges, and units of measurement (beyond the few examples in table footnotes) are not provided.

Table 2 presents several biomarkers without explaining the clinical significance of their abnormal values.

2.3 Data Quality and Handling of Missing Data:

Describe how missing data were addressed, if present.

Specify whether data extraction was performed in duplicate or by trained personnel to ensure accuracy.

Consider mentioning whether blinding was employed during data analysis, as this enhances transparency even in retrospective studies.

3. Statistical Analysis: Limited Interpretation and Reporting

Although basic statistical methods (e.g., Mann-Whitney U test, chi-square test, logistic regression) are appropriately applied, several important elements are underreported or insufficiently interpreted.

3.1 Multivariate Regression:

Report measures of model fit (e.g., the Hosmer-Lemeshow test) and diagnostics for multicollinearity (e.g., Variance Inflation Factor, VIF).

Discuss potential confounders that were evaluated but not included in the final model.

3.2 ROC Analysis:

For the ROC curves presented for AT-III and C4, provide not only the Area Under the Curve (AUC) values and optimal cut-offs but also the corresponding sensitivity, specificity, positive/negative predictive values, and likelihood ratios, which are crucial for assessing clinical utility.

Discuss the clinical rationale or implications of the chosen cut-off points.

3.3 Subgroup Comparisons:

Conduct and report direct comparisons between different types of mixed infections (e.g., bacterial-only vs. viral-only vs. bacterial+viral).

Consider performing stratified analyses by age groups (e.g., <3 years, 3–6 years, >6 years) or by seasonal subsets to explore potential effect modifiers.

4. Results: Presentation and Interpretation

4.1 Strengths:

Clear tabular presentation of demographic data, clinical features, and laboratory results. Appropriate identification of key differences between the single and mixed infection groups.

4.2 Areas for Improvement:

4.2.1 Some tables (e.g., Tables 1 and 3) are densely packed with numbers, percentages, and p-values, making them difficult to interpret quickly. Consider streamlining or using supplementary tables for detailed data.

4.2.2 Several clinically relevant outcomes are not reported in detail, such as: ICU admission rates; Need for and duration of oxygen support; Requirement for mechanical ventilation; Use of adjunctive therapies like corticosteroids or intravenous immunoglobulin (IVIG), which are common in severe *M. pneumoniae* pneumonia.

4.2.3 Some findings lack deeper interpretation. For instance:

Why might gastrointestinal dysfunction and myocardial injury be more common in mixed infections? What could explain the higher frequency of hypokalemia in the single infection group? What is the potential mechanistic or immunological basis for the observed differences in C4 and AT-III levels?

5. Discussion: Needs More Depth and Consideration of Limitations

The Discussion section is somewhat repetitive of the Results, offering limited new synthesis or insight. There is also insufficient discussion on how the study's findings could inform clinical practice or guide antimicrobial stewardship.

Final Recommendation:

Suggest the author to comprehensively revise the above issues and submit them to other magazines.

Response Letter

Dear editor:

Thank you for giving us the opportunity to revise the manuscript. We would like to re-submit the revised manuscript entitled “*Clinical Characteristics, Risk Factors and Pathogen Spectrum of Mixed Infections in Children with Mycoplasma pneumoniae Pneumonia: A Retrospective Cohort Study of 1,428 Cases from a Children's Medical Centre in East China*”. (ID: Spectrum03884-25)

We would like to express our great appreciation to you and reviewers for comments on our paper. Based on the valuable suggestions and comments, the manuscript was revised thoroughly by all the authors, and all corrections were highlighted with red in the manuscript-revised version.

If you have any questions, please feel free to contact us at wyc1116@foxmail.com. Looking forward to hearing from you.

Yours sincerely,

Xin Zhang

January, 17th, 2026

Response to reviewer

Dear reviewer:

Thank you for your careful review and valuable comments concerning on the manuscript. (ID: Spectrum03884-25)

On the basis of those advice, the manuscript was revised thoroughly after the discussion and approval by all the authors, and all correction were highlighted with **red** in the manuscript-revised version. The main corrections in the paper and response to the reviewers are summarized as below.

We would like to express our great appreciation for your suggestions and comments. It is valuable and very helpful for revising and improving the paper, as well as the important guiding significance to our researches.

If you have any questions, please feel free to contact us at wyc1116@foxmail.com. Looking forward to hearing from you.

Yours sincerely,

Xin Zhang

January, 17th, 2026

Response to reviewer #1:

Firstly, please allow me to express our sincerest gratitude. Thank you very much for carefully and patiently reviewing the manuscript and expressing your approval of it. We are so sorry that some of our negligence did cause your misunderstanding. According to your constructive suggestion, the entire text has been modified accordingly. The main corrections in the paper and response to the reviewers are summarized as below.

1. Introduction

Research needs to strengthen the targeting of research gaps. Currently, only "insufficient research on pathogen spectrum and risk factors" is mentioned. Specific comparisons should be made between the pathogen differences of MPP mixed infections in different regions at home and abroad (such as North China, South China, and Europe and America), highlighting the unique epidemiological background of children in East China and clarifying the filling value of this study.

Responses: Thank you for your kind questions to make the manuscript more comprehensive and complete in content. According to your constructive suggestion, the content has been rephrased. (Line 95-117, References)

2. Methods

The "13 plex respiratory pathogen nucleic acid testing" should list all 13 pathogens in the Methods section rather than only in the results tables, to ensure transparency and reproducibility. Classification of influenza viruses: "H3N2" and "H1N1" are listed separately in Table 5 but should be clearly categorized

under "Influenza virus (InfA/InfB)" in the Methods, with a note that they represent subtypes of InfA, to avoid confusion.

Responses: Sorry for the unclear description. The corresponding content has been revised. (Line 173-180, Table 7)

3. Results

3.1 In Table 1, cough duration: Both groups show a median of 6 days, but the interquartile ranges differ (4-9 days for mixed infection vs. 4-7 days for single infection). Please provide the non parametric test statistic (e.g., Z value) in addition to the P value.

Responses: The corresponding $Z/t/\chi^2$ value has been supplemented. (Table 2, Table 3, Table 4, Table 7)

3.2 Bacterial spectrum seasonality: Presenting the annual/seasonal distribution only in a table limits readability. Adding a visual display (e.g., a heatmap, line chart, or stacked bar chart) would help readers grasp trends more intuitively.

Responses: Thank you so much. According to your feedback, the seasonal distribution of common bacteria has been supplemented, as shown in Figure 4. (Line 337-339)

3.3 Complications: There is an apparent calculation error for sepsis in the mixed infection group: $9/900 = 1.00\%$, not 10% . Please thoroughly verify all percentages, counts, and corresponding P values throughout the manuscript.

Responses: Thanks for your advice. Verified all percentages, counts, and corresponding P -values for the entire text. (Table 4)

4. Discussion (The discussion is well written and data driven, but it tends to reiterate results rather than explore underlying mechanisms.)

4.1 Mechanism and literature comparison: Strengthen the comparison with other studies-not only highlighting consistencies but also discussing discrepancies, deepen the mechanistic explanation.

Responses: Thank you for your valuable feedback. We will focus on revising the discussion section by deepening mechanistic explanations, strengthening literature comparisons, and particularly enhancing the exploration of divergent results to reduce result repetition and improve the depth of analysis. (Discussions)

4.2 Regional comparison of pathogen spectrum: Systematically compare your pathogen profile with those reported in other regional studies (both domestic and international) to illustrate whether your findings are region specific or more universal.

Responses: Thank you for raising the important question. We systematically compared the pathogen spectrum of this study with other regional studies. (Line 527-549)

4.3 Move from description to interpretation: Instead of simply stating "we found lower C4 in the mixed infection group," explain why this might occur and what it implies. For example: "The lower complement C4 level in mixed infections suggests excessive consumption of the complement pathway when the host confronts multiple pathogens, which may contribute to immune dysregulation and more severe disease. This aligns with the findings of Andrejeva et al.,

indicating that C4 is not merely a biomarker but potentially an active player in disease progression."

Responses: Thank you for your valuable suggestion. The shift from descriptive statements to mechanistic explanations is crucial for enhancing the scientific value of research. According to your guidance, our modifications are as follows Line 408-422.

4.4 Improve structure: The discussion currently appears as a single, dense block of text. Adding subheadings (e.g., Summary of key findings, Clinical and epidemiological profile, Immunological and coagulation characteristics, Pathogen spectrum and synergy, Clinical implications and future directions) would greatly enhance readability.

Responses: We appreciate your relevant suggestions. Based on your suggestion, we have improved the structure by adding subtitles to enhance readability. (Discussions)

5. Formatting and Stylistic Issues

5.1 Abbreviations: Ensure that all abbreviations are spelled out at first use (e.g., "antithrombin III (AT III)").

Responses: This section has been modified. (Line 42)

5.2 Table consistency: Unify the presentation of data in tables (use "n (%)" throughout) and ensure a space between numbers and units (e.g., "8 days" not "8 days").

Responses: Thanks. This section has been revised and the entire text has been verified.

5.3 References: Check that all references include volume, issue, and page numbers where required, and correct any incomplete entries.

Responses: Thank you for your correction. The corresponding content has been modified. (References)

Response to reviewer #2:

Firstly, please allow me to express our sincerest gratitude. Thank you very much for carefully and patiently reviewing the manuscript and expressing your approval of it. We are so sorry that some of our negligence did cause your misunderstanding. According to your constructive suggestion, the entire text has been modified accordingly. The main corrections in the paper and response to the reviewers are summarized as below.

1. Language, Grammar, and Writing Quality [Needs Major Revision]

The manuscript contains numerous instances of unclear, awkwardly phrased, or grammatically incorrect English. Many sentences are excessively long, lack logical flow, or employ imprecise terminology. These issues significantly compromise the clarity, professionalism, and readability of the study. Recommendations: Professional English language editing is essential. Consider utilizing services from native-speaking scientific editors (e.g., Elsevier Language Services, Editage, or Nature Research Editing Service). Ensure consistent terminology throughout the manuscript (e.g., standardize the use of "mixed infection" versus "co-infection"). Simplify overly complex or redundant sentences to enhance the academic tone and overall clarity.

Responses: Thank you for your valuable feedback on the language and writing quality of the manuscript. We fully agree with your assessment that the current English expression is insufficient and has compromised the clarity and professionalism of the research. We have undertaken comprehensive English polishing and quality improvement throughout the entire manuscript. If necessary, we can make further modifications.

2. Methods Section: Lack of Critical Details. While the methodological approach is generally sound, the omission of several key details affects the study's reproducibility and the assessment of its validity.

2.1 Pathogen Detection Assays: For PCR-based tests (e.g., the 13-plex Respiratory Pathogen Panel), provide information on the specific commercial kit, manufacturer, primer/probe sequences, detection limits, and validated targets. For *Mycoplasma pneumoniae* IgM (MP-IgM) testing, specify the assay's sensitivity and specificity, as well as the reference values used to define positivity. For bacterial culture, detail the colony count thresholds considered positive, quality control procedures for media, and the standards for microbial identification.

Responses: Thank you for your thorough review and valuable suggestions. Providing complete methodological details is crucial for evaluating the quality of the study. As requested, we have now supplemented the detailed information in the “Materials and Methods” section (subsection “2.2 Etiological Detection Methods”) of the manuscript.

(Line 173-200)

2.2 Laboratory Biomarkers:

While numerous biomarkers are listed (e.g., C3, C4, IgA, IgG, AT-III, sCRP, SAA), the methodologies, specific assay kits, reference ranges, and units of measurement (beyond the few examples in table footnotes) are not provided. Table 2 presents several biomarkers without explaining the clinical significance of their abnormal values.

Responses: This section has been modified. (Table 1). These biomarkers (in Table 3, formerly Table 2) mainly include immune function related indicators, coagulation function related indicators and inflammation indicators. The abnormal values of these

markers mainly reflect the infection situation, which is also the content analyzed in this study. Please refer to the discussion section for details.

2.3 Data Quality and Handling of Missing Data: Describe how missing data were addressed, if present. Specify whether data extraction was performed in duplicate or by trained personnel to ensure accuracy. Consider mentioning whether blinding was employed during data analysis, as this enhances transparency even in retrospective studies.

Responses: Thank you for raising this crucial methodological question regarding the handling of missing data. In this retrospective study, data were derived from routine clinical practice, and not all included patients received a uniform, comprehensive panel of laboratory tests. Consequently, for each specific laboratory parameter (e.g., serum amyloid A, complement C4, etc.), the data exhibit varying degrees of incompleteness. When conducting comparisons for any specific indicator between two groups (e.g., single infection group vs. mixed infection group), our analysis was strictly limited to the subset of patients who had valid test results for that particular indicator. The unit of analysis was the "test result," not the "entire enrolled patient cohort." This is a pragmatic and widely adopted approach for handling the common, non-systematic missing data inherent in clinical retrospective research. It ensures that the comparisons are based on actually measured values, thereby avoiding the potential biases that could be introduced by imputing missing values. Other modifications can be found in (Line 209-210).

3. Statistical Analysis: Limited Interpretation and Reporting. Although basic statistical methods (e.g., Mann-Whitney U test, chi-square test, logistic regression)

are appropriately applied, several important elements are underreported or insufficiently interpreted.

3.1 Multivariate Regression:

Report measures of model fit (e.g., the Hosmer-Lemeshow test) and diagnostics for multicollinearity (e.g., Variance Inflation Factor, VIF).

Discuss potential confounders that were evaluated but not included in the final model.

Responses: Thank you for your detailed comments on the multivariate regression analysis section. These suggestions are greatly helpful in enhancing the methodological rigor and transparency of our study. In this study, the multivariate regression analysis only included variables that were significant in the univariate analysis.

3.2 ROC Analysis:

For the ROC curves presented for AT-III and C4, provide not only the Area Under the Curve (AUC) values and optimal cut-offs but also the corresponding sensitivity, specificity, positive/negative predictive values, and likelihood ratios, which are crucial for assessing clinical utility. Discuss the clinical rationale or implications of the chosen cut-off points.

Responses: The modifications have been highlighted throughout the text. Thank you for helping us significantly improve the rigor and completeness of our paper. (Line 282-287, Table 6)

3.3 Subgroup Comparisons: Conduct and report direct comparisons between different types of mixed infections (e.g., bacterial-only vs. viral-only vs.

bacterial+viral). Consider performing stratified analyses by age groups (e.g., <3 years, 3-6 years, >6 years) or by seasonal subsets to explore potential effect modifiers.

Responses: A comparison between different types of mixed infections has been reported, please refer to the supplementary materials for details. (*Supplementary materials Table S1-S6*). Subsequent research will consider stratified analysis by age group (e.g.<3 years, 3-6 years,>6 years) or by season to explore potential effect modifying factors.

4. Results: Presentation and Interpretation

4.2.1 Some tables (e.g., Tables 1 and 3) are densely packed with numbers, percentages, and p-values, making them difficult to interpret quickly. Consider streamlining or using supplementary tables for detailed data.

Responses: Thank you for your valuable suggestions on the structure and depth of the paper. Regarding table readability: Bold all parts of the table with P-values less than 0.05 and mark "*" (Bold column of P-values representatives meaningful data)" below the table. (*Table 2-4, Table 7*)

4.2.2 Several clinically relevant outcomes are not reported in detail, such as:ICU admission rates; Need for and duration of oxygen support; Requirement for mechanical ventilation; Use of adjunctive therapies like corticosteroids or intravenous immunoglobulin (IVIG), which are common in severe M. pneumoniae pneumonia.

Responses: Some indicators include ICU occupancy rate, whether oxygen therapy is needed and its duration; Is mechanical ventilation necessary; Whether to use adjuvant

therapy such as corticosteroids or intravenous immunoglobulin (IVIG) is common in severe *Mycoplasma pneumoniae* pneumonia, and we consider including it in the subsequent analysis of severe *Mycoplasma pneumoniae* pneumonia.

4.2.3 Some findings lack deeper interpretation.

For instance: Why might gastrointestinal dysfunction and myocardial injury be more common in mixed infections? What could explain the higher frequency of hypokalemia in the single infection group? What is the potential mechanistic or immunological basis for the observed differences in C4 and AT-III levels?

Responses: This section has been modified. (Line 408-422, 446-455, 488-507, 516-524)

5. Discussion: Needs More Depth and Consideration of Limitations

The Discussion section is somewhat repetitive of the Results, offering limited new synthesis or insight. There is also insufficient discussion on how the study's findings could inform clinical practice or guide antimicrobial stewardship.

Responses: Thank you for your insightful comments on the discussion section. The issue you pointed out is very relevant and crucial, namely the lack of comprehensive explanation, clinical relevance, and reflection on local limitations in the current discussion. This is indeed a weak link in our initial draft. We will focus on revising the discussion section, deepening the mechanical explanation, strengthening literature comparison, and especially enhancing the exploration of divergent results to reduce result duplication and improve analysis depth. (Line 596-602, 611-620, Discussions, References)

Re: Spectrum03884-25R1-A (**Clinical Characteristics, Risk Factors and Pathogen Spectrum of Mixed Infections in Children with *Mycoplasma pneumoniae* Pneumonia: A Retrospective Cohort Study of 1,428 Cases from a Children's Medical Centre in East China**)

Dear Ms. Xin Zhang:

Thank you for the privilege of reviewing your work. Below you will find my comments, instructions from the Spectrum editorial office, and the reviewer comments.

Reviewer 2

This manuscript presents a well-conducted, large-scale retrospective study investigating the important clinical problem of mixed infections in pediatric *Mycoplasma pneumoniae* pneumonia (MPP). The study is methodologically sound, the statistical analysis appears rigorous, and the findings regarding the high prevalence (63.03%), distinct clinical and laboratory profiles, and independent risk factors for mixed infections are significant and clinically relevant.

Major Comments:

Apparent Paradox in Clinical Outcomes: The data show that the mixed infection group had a significantly shorter hospital stay (8 vs. 9 days, $p < 0.001$) yet a higher incidence of severe pneumonia (30.44% vs. 20.27%, $p < 0.001$). This seemingly contradictory finding is not adequately discussed. Please provide a plausible explanation in the Discussion section. Potential reasons could include earlier onset of severe symptoms leading to prompt hospitalization, differences in treatment protocols or response for mixed infections, or variations in discharge criteria. Clarifying this point is crucial for a coherent interpretation of the disease burden.

Pathophysiological Rationale for Key Biomarkers: The identification of decreased complement C4 and elevated antithrombin III (AT-III) as independent protective and risk factors, respectively, is interesting. However, the discussion on their biological plausibility in the context of mixed infections is relatively brief. Please expand the discussion to more explicitly link these findings to the underlying immunology and coagulation dynamics. For C4, elaborate on how complement consumption or dysregulation might predispose to or result from co-infection. For AT-III, discuss its role as an acute-phase reactant and its potential elevation as a compensatory anti-inflammatory or anticoagulant response in mixed infections, referencing relevant literature.

Limitations and Generalizability: The single-center design from Suzhou, East China, is appropriately noted as a limitation. To better contextualize the findings, please briefly discuss the potential implications of this limitation. For instance, are the epidemiological patterns (e.g., autumn peak, dominant pathogens like HRV and *S. pneumoniae*) likely representative of other regions in China or other countries with similar climates/seasonality? Acknowledging this will help readers gauge the external validity of the results.

Minor Comments:

Clarity on Bacterial Seasonality: The viral pathogen spectrum is nicely analyzed for seasonal variation (Table 5). Please consider adding a sentence or two in the Results or Discussion regarding whether any seasonal trends were observed for the main bacterial pathogens (*S. pneumoniae*, *H. influenzae*), as this would provide a more complete epidemiological picture.

Prediction Model Suggestion: The conclusion recommends developing a prediction model. Since the ROC analysis for C4 and AT-III shows modest AUCs (0.617 and 0.570), it would be helpful to state more explicitly that these two biomarkers alone have limited discriminatory power and that a future multivariable model integrating clinical factors (age, season, symptoms) is needed for robust prediction. This sets a clearer direction for subsequent research.

Reviewer 3

Major Concerns and Limitations

- **Retrospective Single-Center Design:** This is acknowledged by the authors but warrants greater emphasis regarding its limitations, particularly selection bias and limited generalizability of the pathogen spectrum and risk factors to other geographic regions or healthcare settings.
- **Data Presentation and Statistical Analysis:**
 - **Table 3 (Laboratory indicators):** The presentation is confusing. Some results are presented as Mean {plus minus} SD while others as Median (IQR), but the tests used (t-test vs. Mann-Whitney U) are not explicitly stated for each variable in the table. A footnote clarifying the test for each data type is essential. Furthermore, the P-value for Fib is listed as < 0.001 , but the Z-score is -0.264, which seems inconsistent and must be verified.
 - **Multivariate Regression (Table 5):** The result for the C4 variable is alarming. The Odds Ratio (OR) is reported as 0.000005

(6.9727 E9-0.003). An OR this close to zero with an enormous confidence interval suggests severe statistical issues, likely complete separation or multicollinearity. This makes the model unstable and the result for C4 unreliable. The authors must thoroughly check their model assumptions and variable coding. It may be necessary to categorize continuous variables like C4 rather than using them as linear terms.

◦ ROC Analysis (Table 6): The Area Under the Curve (AUC) for both AT-III (0.570) and C4 (0.617) indicates very poor to poor discriminative ability. The claim that they have "moderate predictive value" is an overstatement. The conclusion should be reframed to state that while these factors are independently associated, their predictive power as standalone tests is low, reinforcing the need for a multi-parameter model.

• Interpretation of Results:

◦ The shorter hospital stay in the mixed infection group (8 vs. 9 days, $P < 0.001$) is highlighted but not sufficiently discussed. Is this clinically meaningful? Could it be related to earlier intervention with broader antibiotics or differences in discharge criteria? This counterintuitive finding needs a deeper exploration.

◦ The discussion on inflammatory markers (lower SAA, sCRP in mixed infections) is interesting but speculative. The explanation involving viral suppression of inflammation is one hypothesis, but other factors (e.g., timing of sample collection) could be equally important and should be mentioned as alternative explanations.

Revision Guidelines

Sincerely,
Ping Ren
Editor
Microbiology Spectrum

Overall Recommendation: Reject or require significant revisions

The manuscript presents a valuable retrospective analysis of a large cohort of pediatric MPP patients, highlighting the significant issue of mixed infections. The study is well-motivated and addresses a clinically relevant topic. However, the manuscript requires significant revisions to improve clarity, statistical rigor, and the interpretation of results before it can be considered for publication. The major strengths are the large sample size and comprehensive pathogen detection. The major weaknesses lie in the data presentation, methodological details, and some overinterpretations of the findings.

1. Strengths

- **Large and Well-Defined Cohort:** The study includes a substantial number of participants (n=1,428), which provides considerable statistical power for the analyses performed.
- **Comprehensive Pathogen Detection:** The use of a 13-plex PCR panel alongside bacterial culture and serology allows for a detailed assessment

of the mixed infection pathogen spectrum, which is a key contribution of the work.

- **Clinical Relevance:** The focus on risk factors and potential biomarkers (C4, AT-III) for mixed infections has direct implications for improving clinical management and early intervention strategies in pediatric pneumonia.

2. Major Concerns and Limitations

- **Retrospective Single-Center Design:** This is acknowledged by the authors but warrants greater emphasis regarding its limitations, particularly selection bias and limited generalizability of the pathogen spectrum and risk factors to other geographic regions or healthcare settings.
- **Data Presentation and Statistical Analysis:**
 - **Table 3 (Laboratory indicators):** The presentation is confusing. Some results are presented as Mean \pm SD while others as Median (IQR), but the tests used (t-test vs. Mann-Whitney U) are not explicitly stated for each variable in the table. A footnote clarifying the test for each data

type is essential. Furthermore, the P-value for Fib is listed as <0.001 , but the Z-score is -0.264 , which seems inconsistent and must be verified.

- Multivariate Regression (Table 5): The result for the C4 variable is alarming. The Odds Ratio (OR) is reported as 0.000005 ($6.9727 \text{ E}9-0.003$).

An OR this close to zero with an enormous confidence interval suggests severe statistical issues, likely complete separation or multicollinearity.

This makes the model unstable and the result for C4 unreliable. The authors must thoroughly check their model assumptions and variable coding. It may be necessary to categorize continuous variables like C4 rather than using them as linear terms.

- ROC Analysis (Table 6): The Area Under the Curve (AUC) for both AT-III (0.570) and C4 (0.617) indicates very poor to poor discriminative ability. The claim that they have "moderate predictive value" is an overstatement. The conclusion should be reframed to state that while these factors are independently associated, their predictive power as standalone tests is low, reinforcing the need for a multi-parameter model.

- Interpretation of Results:

- The shorter hospital stay in the mixed infection group (8 vs. 9 days, $P < 0.001$) is highlighted but not sufficiently discussed. Is this clinically meaningful? Could it be related to earlier intervention with broader antibiotics or differences in discharge criteria? This counterintuitive finding needs a deeper exploration.

- The discussion on inflammatory markers (lower SAA, sCRP in mixed infections) is interesting but speculative. The explanation involving viral suppression of inflammation is one hypothesis, but other factors (e.g., timing of sample collection) could be equally important and should be mentioned as alternative explanations.

Response Letter

Dear editor:

Thank you for giving us the opportunity to revise the manuscript. We would like to re-submit the revised manuscript entitled “*Clinical Characteristics, Risk Factors and Pathogen Spectrum of Mixed Infections in Children with Mycoplasma pneumoniae Pneumonia: A Retrospective Cohort Study of 1,428 Cases from a Children's Medical Centre in East China*”. (ID: Spectrum03884-25R1-A)

We would like to express our great appreciation to you and reviewers for comments on our paper. Based on the valuable suggestions and comments, the manuscript was revised thoroughly by all the authors, and all corrections were highlighted with red in the manuscript-revised version.

If you have any questions, please feel free to contact us at wyc1116@foxmail.com. Looking forward to hearing from you.

Yours sincerely,

Xin Zhang

March, 31th, 2026

Response to reviewer

Dear reviewer:

Thank you for your careful review and valuable comments concerning the manuscript. (ID: Spectrum03884-25R1-A)

On the basis of those advice, the manuscript was revised thoroughly after the discussion and approval by all the authors, and corrections were highlighted with red in the manuscript-revised version. The main corrections in the paper and response to the reviewers are summarized as below.

We would like to express our great appreciation for your suggestions and comments. It is valuable and very helpful for revising and improving the paper, as well as the important guiding significance to our researches.

If you have any questions, please feel free to contact us at wyc1116@foxmail.com. Looking forward to hearing from you.

Yours sincerely,

Xin Zhang

March, 31th, 2026

Response to reviewer #2:

Firstly, please allow me to express our sincerest gratitude. Thank you very much for patiently reviewing the manuscript and expressing your approval of it. We have carefully considered all the points raised and have revised the manuscript accordingly. The main corrections in the paper and response to the reviewers are summarized as below. Meanwhile, to improve clarity and enhance readability, we have revised the manuscript through the AJE website, which specializes in editing and proofreading scientific manuscripts, ensuring that the English language, grammar, punctuation, spelling, and overall style of this manuscript are further refined. (Verification code: 2A5F-9A24-8846-4405-8578; this can be verified on the AJE website).

Major Comments:

1. Apparent Paradox in Clinical Outcomes: The data show that the mixed infection group had a significantly shorter hospital stay (8 vs. 9 days, $p < 0.001$) yet a higher incidence of severe pneumonia (30.44% vs. 20.27%, $p < 0.001$). This seemingly contradictory finding is not adequately discussed. Please provide a plausible explanation in the Discussion section. Potential reasons could include earlier onset of severe symptoms leading to prompt hospitalization, differences in treatment protocols or response for mixed infections, or variations in discharge criteria. Clarifying this point is crucial for a coherent interpretation of the disease burden.

Response: Thank you for your kind questions to make the manuscript more comprehensive and complete in content. We agree with your suggestion and have held an in-depth discussion accordingly. In the revised manuscript, we have added some possible explanations. Details of revisions are shown in (Line 404-416).

2. Pathophysiological Rationale for Key Biomarkers: The identification of decreased complement C4 and elevated antithrombin III (AT-III) as independent risk factors, respectively, is interesting. However, the discussion on their biological plausibility in the context of mixed infections is relatively brief. Please expand the discussion to more explicitly link these findings to the underlying immunology and coagulation dynamics. For C4, elaborate on how complement consumption or dysregulation might predispose to or result from co-infection. For AT-III, discuss its role as an acute-phase reactant and its potential elevation as a compensatory anti-inflammatory or anticoagulant response in mixed infections, referencing relevant literature.

Response: We greatly appreciate your constructive and insightful comments on the discussion of pathophysiological mechanisms and biological plausibility of complement C4 and AT-III in mixed infections. We have carefully revised the manuscript in accordance with your suggestions. After reviewing relevant literature, we have strengthened the mechanistic discussion, and more explicitly linked our findings to potential immunological and coagulation dynamics. All revisions in the manuscript have been highlighted in red for your easy review. (Line 448-460, 484-494, Reference 32, 33, 39, 40).

3. Limitations and Generalizability: The single-center design from Suzhou, East China, is appropriately noted as a limitation. To better contextualize the findings, please briefly discuss the potential implications of this limitation. For instance, are the epidemiological patterns (e.g., autumn peak, dominant pathogens like HRV and *S. pneumoniae*) likely representative of other regions in China or other

countries with similar climates/seasonality? Acknowledging this will help readers gauge the external validity of the results.

Response: We greatly appreciate your constructive and valuable comment. In response to your comment, we have added a paragraph regarding the generalizability of our findings in the limitations section (Lines 690-697) of the manuscript. In addition, corresponding revisions have been made in the Materials and Methods section. (Lines 148-152).

Minor Comments:

1. Clarity on Bacterial Seasonality: The viral pathogen spectrum is nicely analyzed for seasonal variation (Table 5). Please consider adding a sentence or two in the Results or Discussion regarding whether any seasonal trends were observed for the main bacterial pathogens (*S. pneumoniae*, *H. influenzae*), as this would provide a more complete epidemiological picture.

Response: Thank you so much. We have supplemented the description of the seasonal distribution of major bacterial pathogens in the Results and Discussions sections. This supplement has improved the epidemiological profile of the present study. (Line 358-360, 660-674).

2. Prediction Model Suggestion: The conclusion recommends developing a prediction model. Since the ROC analysis for C4 and AT-III shows modest AUCs (0.617 and 0.570), it would be helpful to state more explicitly that these two biomarkers alone have limited discriminatory power and that a future multivariable model integrating clinical factors (age, season, symptoms) is

needed for robust prediction. This sets a clearer direction for subsequent research.

Response: Thank you for your constructive suggestion on the prediction model. In response to your comment, we have revised this section accordingly. Details of revisions are shown in (Lines 553-568).

Response to reviewer #3:

Firstly, please allow me to express our sincerest gratitude. Thank you very much for patiently reviewing the manuscript and expressing your approval of it. We have carefully considered all the points raised and have revised the manuscript accordingly. The main corrections in the paper and response to the reviewers are summarized as below. Meanwhile, to improve clarity and enhance readability, we have revised the manuscript through the AJE website, which specializes in editing and proofreading scientific manuscripts, ensuring that the English language, grammar, punctuation, spelling, and overall style of this manuscript are further refined. (Verification code: 2A5F-9A24-8846-4405-8578; this can be verified on the AJE website).

Major Concerns and Limitations

1. Retrospective Single-Center Design: This is acknowledged by the authors but warrants greater emphasis regarding its limitations, particularly selection bias and limited generalizability of the pathogen spectrum and risk factors to other geographic regions or healthcare settings.

Response: We greatly appreciate your constructive and valuable comment. In response to your comment, we have added a paragraph regarding the generalizability of our findings in the limitations section (Lines 690-697) of the manuscript. In addition, corresponding revisions have been made in the Materials and Methods section (Lines 148-152).

2. Data Presentation and Statistical Analysis

2.1 Table 3 (Laboratory indicators): The presentation is confusing. Some results are presented as Mean {plus minus} SD while others as Median (IQR), but the

tests used (t-test vs. Mann-Whitney U) are not explicitly stated for each variable in the table. A footnote clarifying the test for each data type is essential. Furthermore, the P-value for Fib is listed as <0.001, but the Z-score is -0.264, which seems inconsistent and must be verified.

Response: Thank you for your thorough review and valuable suggestions. We have revised Table 3 and added footnotes specifying the statistical tests used for each variable. Meanwhile, we have also added footnotes to Table 2. In addition, we have rechecked all tables throughout the manuscript, especially the data of Fib. The previously reported Z-score of -0.264 was found to be a typographical error. The correct Z-score and corresponding *P*-value are now accurately reported in the revised Table 3. (Table2, Table3, Line 248-251, 266-269).

2.2 Multivariate Regression (Table 5): The result for the C4 variable is alarming. The Odds Ratio (OR) is reported as 0.000005 (6.9727 E9-0.003). An OR this close to zero with an enormous confidence interval suggests severe statistical issues, likely complete separation or multicollinearity. This makes the model unstable and the result for C4 unreliable. The authors must thoroughly check their model assumptions and variable coding. It may be necessary to categorize continuous variables like C4 rather than using them as linear terms.

Response: Thank you very much for your valuable comments and rigorous guidance regarding the multivariate regression analysis section of this study. We highly value your observation that the odds ratio for the C4 variable in the original Table 5 was 0.000005, with an unusually wide confidence interval. We have conducted a comprehensive review and in-depth analysis. After re-examination, we confirmed that this issue was primarily due to unmet model assumptions and inappropriate variable

handling. To address this issue, we categorized C4 into quartile-based groups: low-value group Q1 ($<P_{25}$), medium-low-value group Q2 ($P_{25}-P_{50}$), medium-high-value group Q3 ($P_{50}-P_{75}$), and high-value group Q4 ($>P_{75}$). The logistic regression analysis was repeated, and after controlling for confounding factors, the results showed that a C4 level in the high-value group Q4 was an independent factor distinguishing between single MP infection and mixed infection, with a statistically significant OR of 0.02 (95% CI: 0.002-0.246, $P<0.05$). We have updated the relevant data in Table 5 in the revised manuscript, supplemented the related discussion, and added details on the handling of the C4 variable in the Materials and Methods section. Once again, we appreciate your rigorous review and valuable suggestions, which have helped us identify deficiencies in our study and significantly improved its quality. We have made the corresponding revisions as described above and kindly request your further review. (Table5, Lines 221-223, 283-296, 545-553).

2.3 ROC Analysis (Table 6): The Area Under the Curve (AUC) for both AT-III (0.570) and C4 (0.617) indicates very poor to poor discriminative ability. The claim that they have "moderate predictive value" is an overstatement. The conclusion should be reframed to state that while these factors are independently associated, their predictive power as standalone tests is low, reinforcing the need for a multi-parameter model.

Response: Thank you for your valuable feedback. We fully agree with your comment that describing C4 and AT-III as having “moderate predictive value” in the original manuscript was not sufficiently rigorous. In response to your advice, we have revised the relevant description to clearly state that the AUC values of AT-III (0.570) and C4

(0.617) indicate low predictive efficacy and unsatisfactory accuracy when used as individual indicators. Details of revisions are shown in (Lines 553-568).

3. Interpretation of Results:

3.1 The shorter hospital stay in the mixed infection group (8 vs. 9 days, $P < 0.001$) is highlighted but not sufficiently discussed. Is this clinically meaningful? Could it be related to earlier intervention with broader antibiotics or differences in discharge criteria? This counterintuitive finding needs a deeper exploration.

Response: Thank you for your kind questions to make the manuscript more comprehensive and complete in content. We agree with your suggestion and have held an in-depth discussion accordingly. In the revised manuscript, we have added some possible explanations. Details of revisions are shown in (Line 404-416).

3.2 The discussion on inflammatory markers (lower SAA, sCRP in mixed infections) is interesting but speculative. The explanation involving viral suppression of inflammation is one hypothesis, but other factors (e.g., timing of sample collection) could be equally important and should be mentioned as alternative explanations.

Response: Thank you for your insightful comment. We agree that the discussion on the lower levels of inflammatory markers (SAA and sCRP) in mixed infections is somewhat speculative, and the explanation involving viral suppression of the inflammatory response should be presented more cautiously as a hypothesis rather than a definitive conclusion. In the revised manuscript, we have added alternative explanations, including the potential influence of the timing of sample collection, as

well as the effects of the patients' baseline immune status and early administration of antiviral or antibacterial agents. Details of revisions are shown in (Line 504-525).

Re: Spectrum03884-25R2 (**Clinical Characteristics, Risk Factors and Pathogen Spectrum of Mixed Infections in Children with *Mycoplasma pneumoniae* Pneumonia: A Retrospective Cohort Study of 1,428 Cases from a Children's Medical Centre in East China**)

Dear Ms. Xin Zhang:

Your manuscript has been accepted, and I am forwarding it to the ASM production staff for publication. Your paper will first be checked to make sure all elements meet the technical requirements. ASM staff will contact you if anything needs to be revised before copyediting and production can begin. Otherwise, you will be notified when your proofs are ready to be viewed.

Sincerely,
Ping Ren
Editor
Microbiology Spectrum